# Predictive and robust gene selection for spatial transcriptomics

Ian Covert[1], Rohan Gala [2], Tim Wang[3], Karel Svoboda [3,4], Uygar Sümbül [2] ✉ & Su-In Lee [1] ✉

A prominent trend in single-cell transcriptomics is providing spatial context alongside a characterization of each cell's molecular state. This typically requires targeting an a priori selection of genes, often covering less than 1% of the genome, and a key question is how to optimally determine the small gene panel. We address this challenge by introducing a flexible deep learning framework, PERSIST, to identify informative gene targets for spatial transcriptomics studies by leveraging reference scRNA-seq data. Using datasets spanning different brain regions, species, and scRNA-seq technologies, we show that PERSIST reliably identifies panels that provide more accurate prediction of the genome-wide expression profile, thereby capturing more information with fewer genes. PERSIST can be adapted to specific biological goals, and we demonstrate that PERSIST's binarization of gene expression levels enables models trained on scRNA-seq data to generalize with to spatial transcriptomics data, despite the complex shift between these technologies.

Cell type classification has been revolutionized by recent advances in single-cell genomics. It is now possible to profile the entire mRNA repertoire (i.e., the transcriptome) of tens, or even hundreds, of thousands of individual cells (scRNA-seq) in a single experiment. Large-scale scRNA-seq studies have provided high-resolution taxonomies of the transcriptomic cell types in many tissues across several species, and leveraging data from these scRNA-seq studies, spatial transcriptomic methods can examine molecularly defined cells in their native context[1–10]. This has led to their use in large consortia such as the Human Cell Atlas and Brain Initiative Cell Census Network, and widespread recognition of their promise[11].

In particular, fluorescence in situ hybridization (FISH) is a prominent spatial transcriptomics approach, and it is the basis of many recently developed technologies[3,4,6,7,12]. By revealing the spatial organization of cells within tissues, FISH is playing a central role in uncovering the fundamental principles of brain organization[12]. In conjunction with other experimental modalities (e.g., morphological, connectivity, or electrophysiological studies), FISH can also link transcriptomic identity with other data to provide a better understanding of the functional role of individual cell types[9,13–16].

Whereas scRNA-seq detects genes in a largely unbiased manner, FISH-based technologies assay a pre-defined list of genes. In routine FISH experiments, only a small fraction of the transcriptome is targeted[3,6,9,12]; this is in part because the complexity and duration of FISH experiments increases sharply with the number of target genes, and also because highly specialized methods capable of probing thousands of genes are applicable only to thin tissue sections and cultured cells[7,17]. Thus, judicious selection of a small number of highly informative target genes (i.e., a gene panel) is key for most FISH experiments. Experimentalists often rely on ad hoc approaches to gene selection, most commonly choosing markers based on prior knowledge or very high expression in a limited subset of cells[9,18]. Such methods are suboptimal as they tend to overlook genes with more complex expression patterns and rarely account for correlated expression between selected genes, which yields redundant information. Here, we frame the identification of markers as a feature selection problem and seek to address it in a principled manner using tools from machine learning.

Feature selection problems arise in many domains[19–23], but spatial transcriptomics studies present unique challenges and thus demand a

[1]Paul G. Allen School of Computer Science & Engineering, University of Washington, Seattle, WA, USA. [2]Allen Institute for Brain Science, Seattle, WA, USA. [3]HHMI Janelia Research Campus, Ashburn, VA, USA. [4]Allen Institute for Neural Dynamics, Seattle, WA, USA. ✉e-mail: uygars@alleninstitute.org; suinlee@cs.washington.edu

specialized solution. Importantly, because reference datasets consisting of spatially resolved mRNA detection measurements for thousands of genes are unavailable, scRNA-seq datasets are used instead to guide the selection process. The use of a surrogate dataset presents new obstacles: the expression counts observed by scRNA-seq and spatial transcriptomics technologies may differ significantly, with a relationship that is nonlinear and noisy[24–26]. Hence, a gene panel selected without considering the difference between the datasets is unlikely to perform as well as intended, as we demonstrate with several existing methods.

Moreover, the optimal gene panel should account for specifics of the target experiment, which may demand tuning of the selection criterion. For instance, linking of spatial characterization of gene expression through spatial transcriptomics and the electrical properties of neurons[14,15] may require a gene panel that relates to membrane excitability. Another scenario could involve investigating a specific subclass of cells; for example, exploring neurons expressing a specific marker gene in a particular brain region demands a gene panel based on reference data from this molecularly and spatially constrained population. Furthermore, certain spatial transcriptomics methods might require the target genes to have either relatively high or low expression levels; for example, when using low-resolution detection methods, it may be preferable to prioritize highly expressed genes.

We address these challenges by introducing PredictivE and Robust gene SelectIon for Spatial Transcriptomics (PERSIST), an algorithm to select genes that can serve as valuable targets in spatial transcriptomics studies. PERSIST uses scRNA-seq data and deep learning to find a small number of highly informative genes whose expression can predict the genome-wide expression profile. In doing so, PERSIST trains a reconstruction model with a loss function that accounts for noisy gene dropouts in scRNA-seq; incorporates expert knowledge by pre-selecting or pre-filtering genes; scales to very large datasets using minibatched training; and quantizes gene expression levels to account for the domain shift between scRNA-seq and spatial transcriptomics. Furthermore, our deep learning-based selection mechanism is flexible: by changing the prediction target, PERSIST can also operate in a supervised rather than unsupervised fashion to address specific experimental aims, such as cell type classification or electrophysiological characterization. Our work focuses primarily on FISH-based studies, but many of the challenges identified above are common to a larger class of spatial transcriptomic methods[8,10], thus suggesting broader applicability of our method.

We validate our approach using reference datasets from different technologies (plate-based SmartSeq and droplet-based 10X), multiple brain regions (V1, ALM, MOp) and different species (mouse, human) on classification and reconstruction tasks. We then highlight PERSIST's flexibility and show how to incorporate a different data modality (electrophysiology) with FISH experiments using a large Patch-seq dataset[14]. Finally, we devise an evaluation procedure to showcase the effectiveness of our robust inference approach based on gene quantization using a recent MERFISH dataset[9], which we show allows predictive models to transfer across technologies despite the measurement differences. Through our comprehensive set of experiments and comparisons with other methods, we provide strong evidence that PERSIST can identify valuable gene targets for spatial transcriptomics studies.

## Results
### Selecting genes using deep learning
Given scRNA-seq data from a cell population to be profiled using spatial transcriptomics, PERSIST selects a small panel of genes that can optimally reconstruct the entire scRNA-seq expression profile. Intuitively, such gene panels are useful for a variety of downstream tasks because they sacrifice minimal information. Our approach is inspired by classical dimension-reduction techniques like principal

components analysis (PCA)[27], but PERSIST selects a discrete set of genes rather than finding linear combinations. Additionally, it reconstructs the original data using a non-linear model and with a quality measure more appropriate for scRNA-seq data. PCA measures reconstruction quality using a mean squared error (MSE) loss, which recent work has found to be ill-suited for scRNA-seq[28,29], so PERSIST instead uses a *hurdle loss function* to account for noisy gene dropouts[30]. 'Dropouts' refer to the failure to detect mRNA transcripts due to inefficiencies in cDNA library preparation, which is prevalent when using lower resolution (and typically higher throughput) platforms such as 10X[31–33]. The hurdle loss used by PERSIST therefore involves separately predicting each gene's expression level and whether it is actually expressed, which lets the model explicitly represent dropout noise in its predictions (see Methods).

PERSIST uses a deep learning model with a custom layer designed to pinpoint a small number of useful input features (Fig. 1). This approach is inspired by recent work on *differentiable feature selection*, which enables neural networks to select features using gradient-based optimization[20,34–36]. The selection layer applies a learned binary mask that sparsifies over the course of the optimization process; information initially flows through the model from all genes, but the relevant inputs are gradually reduced down to a user-specified number (Fig. 1B). The model's memory usage can be managed via the minibatch size used for training, and when necessary, the computational cost can be further reduced by performing a preliminary filtering step (see Methods).

By default, PERSIST operates in an *unsupervised* manner by reconstructing the full scRNA-seq expression profile, which removes the need for any labels or manual annotation. However, PERSIST can also operate in a *supervised* manner by incorporating cell-level annotations as the model's prediction target, such as cell type labels or complementary epigenetic data like chromatin accessibility and methylation (Fig. 1A). As we show in our experiments, this gives PERSIST a versatility that is not shared by other methods, and which lets practitioners select genes that are tailored to meet specific biological questions and objectives. While spatial transcriptomics studies often have specific goals like classifying cell types[3,5,7,9], enabling PERSIST to operate in an unsupervised manner is important because reference cell type clusterings are not always available, consensus definitions of cell types are still evolving[37], and focusing on gene expression enables unbiased characterization of complex tissues and specific brain regions.

The goal of our evaluation is to demonstrate that genes identified by PERSIST can serve as valuable targets in spatial transcriptomics studies. Showing this is not straightforward, both due to the cost of running multiple FISH studies with panels selected by different criteria, as well as the difficulty of providing an unbiased comparison through studies conducted on different tissues. We therefore evaluate the PERSIST gene panels by simulating their use in FISH studies, and in particular via prediction tasks that would be of interest to practitioners in such studies. scRNA-seq and FISH have very different noise sources and detection issues, and the number of transcript counts observed by each technology can differ substantially for the same cell, so our simulated prediction tasks binarize gene expression levels. This pre-processing step allows models to transfer across technologies despite the domain shift, as we demonstrate in an experiment with MERFISH data.

Our experiments compare PERSIST to several widely used and state-of-the-art gene selection methods. We tested the Seurat[38] and Cell Ranger[39] gene selection procedures, which are based on per-gene variance and dispersion levels and are implemented in the popular ScanPy package[40]. These methods are designed primarily to reduce computation and inform clustering studies that help determine marker genes, but they are not intended to directly select gene panels; therefore, their performance is not expected to be competitive with

## A  Method workflow

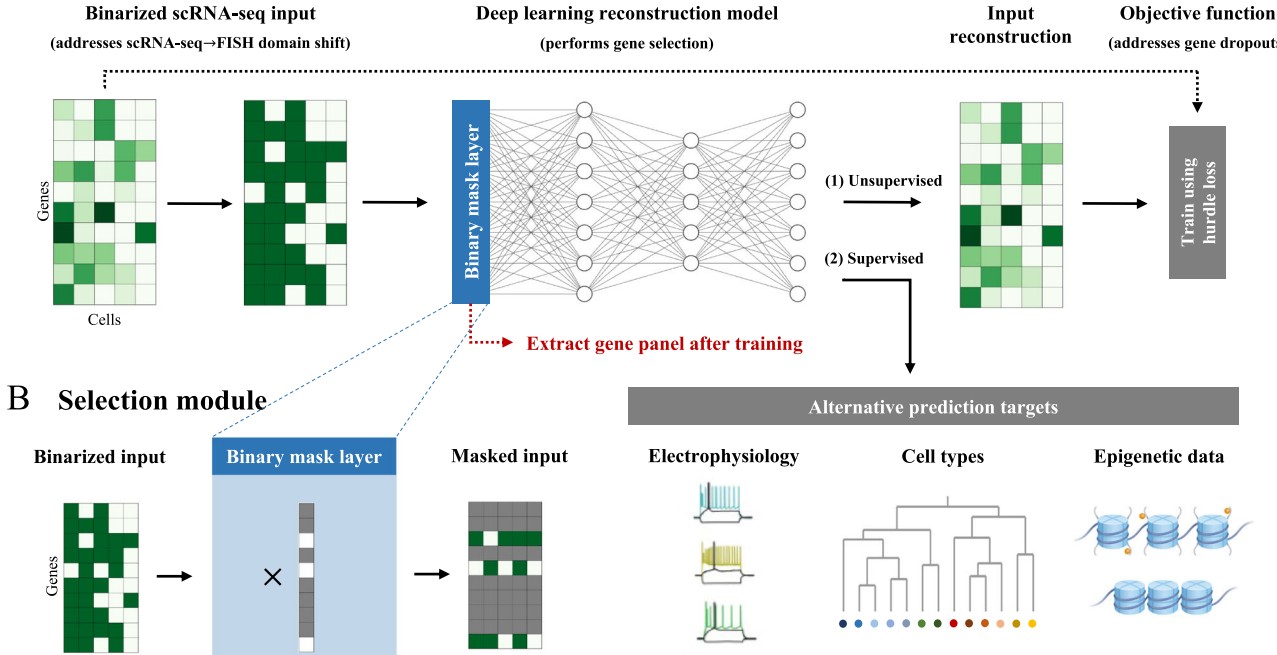

## B  Selection module

**Fig. 1 | Overview of *predictive and robust gene selection for spatial transcriptomics* (PERSIST).** PERSIST selects genes using a deep learning model trained to reconstruct the genome-wide expression profile. **A** The model is trained using scRNA-seq data, which is binarized to address the domain shift relative to FISH measurements. The model by default aims to reconstruct the original scRNA-seq gene expression levels, and the objective function is a hurdle loss designed to account for noisy gene dropouts. Alternatively, one can use a supervised prediction target to address specific experimental aims, such as cell type classification. After training, the selected gene panel is extracted from the model's binary mask layer. **B** The binary mask layer selects genes by multiplying the input with a learned binary mask, which controls the subset of input features that are used to make predictions.

methods designed explicitly for selecting small gene sets. Next, we tested the recently proposed GeneBasis method[41] that selects genes using a greedy algorithm to preserve the data manifold. Finally, we considered three methods that aim to differentiate cell types: a method that maximizes the information about cell type labels (MutInfo)[9,42], a method that identifies key gene predictors using feature importance scores (SMaSH)[43], and the scGeneFit method[44] that uses linear programming. These methods span a range of selection criteria, but PERSIST is a flexible method that can be adapted to multiple experimental objectives relevant to practitioners, and that was designed specifically for transferability to spatial transcriptomics studies.

### PERSIST enables more accurate scRNA-seq expression profile reconstruction

We first tested PERSIST on two scRNA-seq datasets: a SmartSeq v4[45] dataset consisting of 22,160 neurons from the mouse primary visual (V1) and anterior lateral motor (ALM) cortices[46] (hereafter referred to as SSv4), and a 10X[39] dataset consisting of 72,629 neurons from the human motor (M1) cortex[47] (hereafter referred to as 10X). These datasets profile different brain regions and species and were collected using two library preparation platforms that yield different levels of sparsity. Working with an initial set of 10,000 high-variance genes, we used PERSIST and the other gene selection methods to identify panels of 8–256 marker genes, a range that spans the vast majority of FISH studies.

As a benchmark metric for our comparisons, we calculated the portion of variance explained in the genome-wide scRNA-seq expression profile by each selected gene panel. For gene panels of all sizes on both datasets, PERSIST explained more variance and outperformed the unsupervised methods Seurat, Cell Ranger, and GeneBasis (Fig. 2A,

C). The GeneBasis approach is most competitive with PERSIST, but it explained considerably less variance for smaller gene panels, which are most commonly used in experiments. There is diminishing improvement as the panel size increases, and even large panels, such as those with 256 genes, fail to explain all the variance. This is due not only to the many factors of variation in the full expression profiles, but to high noise levels in the data. To verify this, we calculated the amount of variance explained by the cell types in each dataset. We found that cell type labels explained just 19% of the variance in the SSv4 data and 11% in the 10X data, suggesting high intra-type variability due to stochasticity in gene expression and detection. Perhaps surprisingly, the PERSIST panels can explain more variance than the cell type identities given enough genes.

Importantly, PERSIST binarizes gene expression levels during training whereas Seurat, Cell Ranger, and GeneBasis all use either raw or logarithmized expression counts. This creates a degree of inconsistency among methods, so we asked whether this pre-processing step could account for differences in performance. A modified version of GeneBasis becomes more competitive with PERSIST, whereas Seurat and Cell Ranger perform worse with binarization (Supp. Fig. 4). For MutInfo, SMaSH and scGeneFit, which leverage cell type labels to select genes, we find that PERSIST outperforms these methods independent of binarization (Supp. Fig. 3). Overall, the results show that PERSIST's binarization step can be incorporated into several of the baselines to enable better transferability to FISH studies.

Recent work has demonstrated that the "dropout pattern" of a cell (i.e., the set of genes not detected by scRNA-seq) is nearly as informative as quantitative expression levels for identifying cell types[32,48]. We thus wondered whether the selected gene panels could predict the set of genes with non-zero transcript counts in the rest of

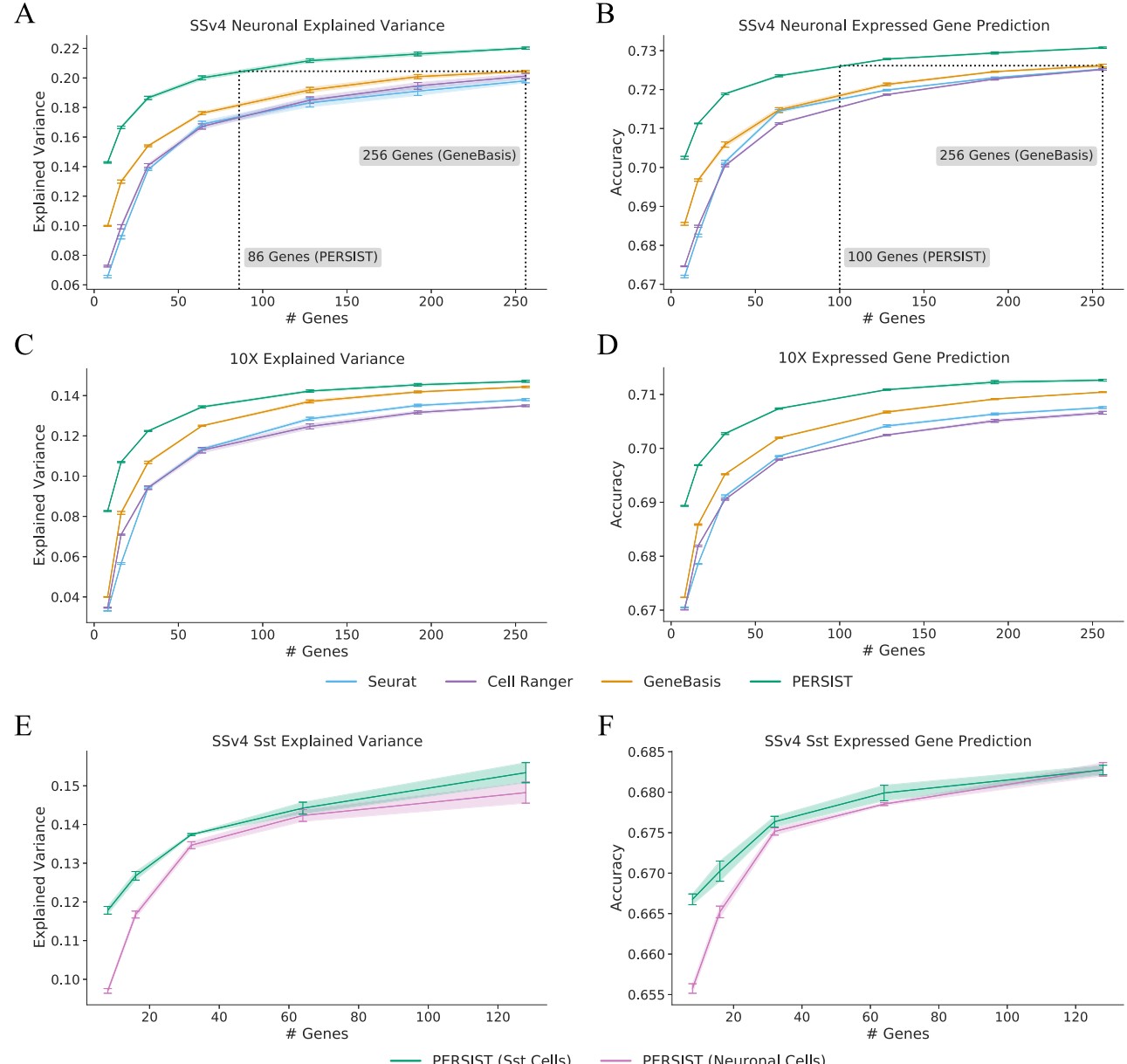

**Fig. 2 | Evaluating gene panels based on their ability to reconstruct scRNA-seq expression profiles.** The selected genes are used to predict either log-normalized expression counts for the remaining genes (**A**, **C**, **E**) or whether each gene is expressed in the genome-wide profile (**B**, **D**, **F**). The panels selected by PERSIST perform better than those selected by previous methods. **A** Explained variance for gene panels identified by PERSIST and other methods with the SSv4 V1/ALM neuronal cells (higher is better). **B** Expressed gene prediction accuracy with the SSv4 V1/ALM neuronal cells (higher is better). **C** Explained variance for gene panels with the 10X M1 cells. **D** Expressed gene prediction accuracy for panels with the 10X M1 cells. **E** Explained variance for gene panels with the SSv4 V1/ALM Sst cells. **F**, Expressed gene prediction accuracy for panels with the SSv4 V1/ALM Sst cells. The results were calculated using a set of held-out cells from each dataset, with $n = 2216$ for the SSv4 neuronal cells (**A**, **B**), $n = 7262$ for the 10X cells (**C**, **D**), and $n = 270$ for the SSv4 Sst cells (**E**, **F**). All error bars represent 95% confidence intervals determined by training with five bootstrapped datasets.

the scRNA-seq dataset (Fig. 2B, D). In this task, PERSIST again performs more accurately than other methods on both datasets for all panel sizes; this is due in part to our hurdle loss function, which involves predicting whether each gene is detected. For this analysis, we excluded genes that were rarely expressed and housekeeping genes that are ubiquitously expressed, focusing on those expressed in 20–80% of cells ($n = 4972$ genes), but similar results were found for varying cutoffs (Supp. Fig. 6). Supp. Fig. 5 shows the prediction accuracy for each gene, revealing that those with moderate mean expression are more difficult to predict than those with predominantly high or low expression.

Finally, from an experimental standpoint, it would be most practical to select a single general-purpose gene panel using the entire dataset rather than generating a gene panel for each particular subtype of interest. To assess the feasibility of this strategy, we focused on the class of somatostatin-expressing (Sst) interneurons (2701 cells) in the SSv4 dataset and compared gene sets selected by PERSIST when trained on the entire scRNA-seq dataset versus just the Sst subpopulation. Performance improved when we only used data from the specific cell type of interest during gene selection (Fig. 2E, F). However, the improvement diminished when 32 or more genes were selected, which suggests that general purpose gene panels may be appropriate

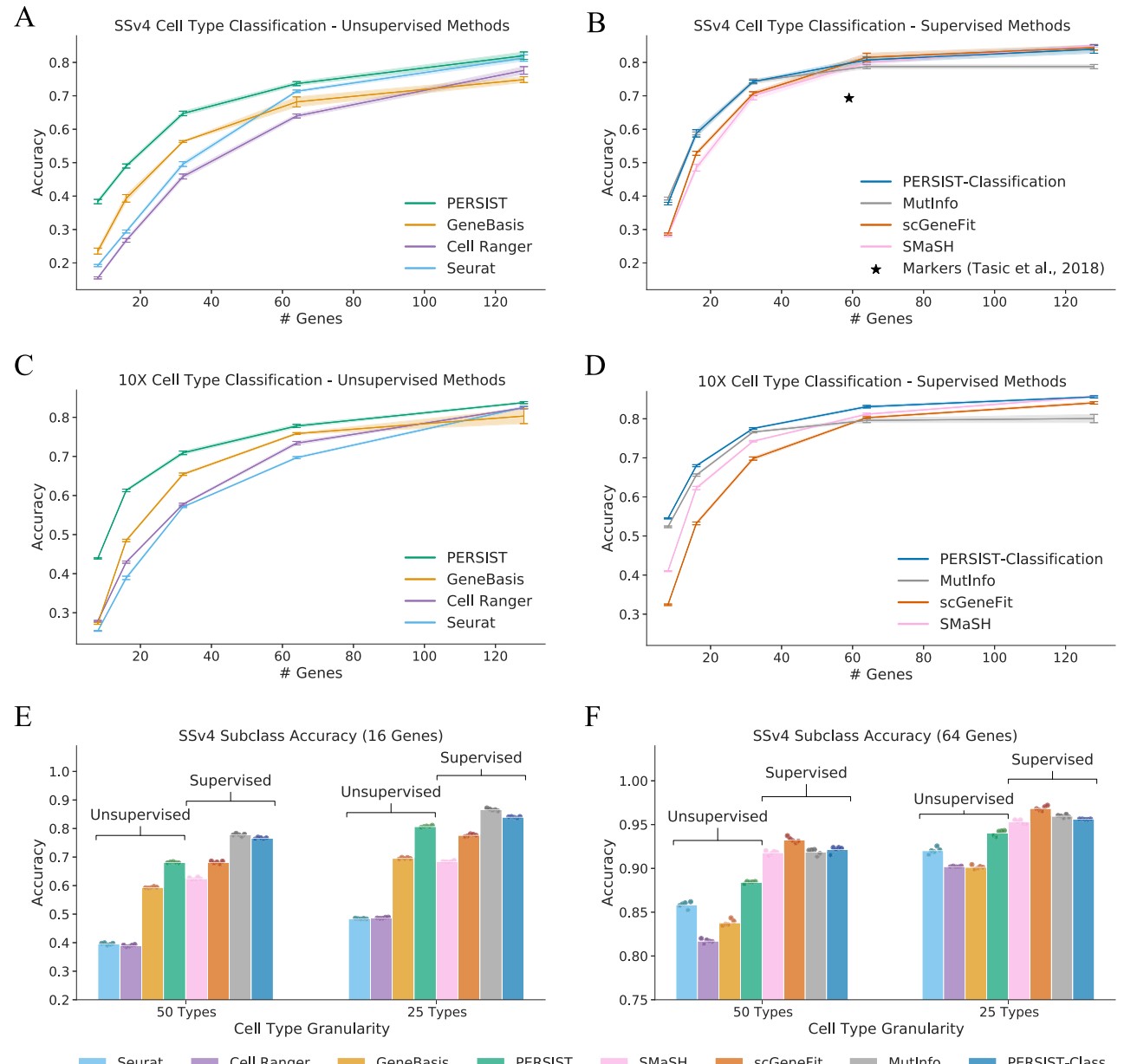

**Fig. 3 | Evaluating gene panels based on their ability to classify cell types.** The selected genes are used to classify scRNA-seq cell types, and PERSIST achieves strong performance despite not using cell type labels during the selection process. PERSIST-Classification, a version modified to distinguish cell types, matches the best existing supervised approaches. **A** Cell type classification accuracy for gene panels selected by unsupervised methods with the SSv4 dataset (higher is better). **B** Accuracy for gene panels selected by supervised methods with the SSv4 dataset. **C** Accuracy for unsupervised methods with the 10X dataset. **D** Accuracy for supervised methods with the 10X dataset. **E** Accuracy for cell subtypes obtained by merging the SSv4 dataset's transcriptomic hierarchy, for panels of 16 genes. **F**, Accuracy with merged SSv4 cell subtypes for panels of 64 genes. The results for each dataset were calculated using a set of held-out cells, with $n = 2216$ for the SSv4 cells (**A**, **B**, **E**, **F**) and $n = 7262$ for the 10X cells (**C**, **D**). All error bars represent 95% confidence intervals from training with five bootstrapped datasets, and **E**, **F** show results from the five runs.

---

for technologies that assay large numbers of genes (e.g., multiplexed methods like MERFISH[4]).

## PERSIST enables accurate cell type classification

As another evaluation metric, we tested how accurately the gene panels selected by each method can classify cell types, which is a common goal of spatial transcriptomics studies[3,5,7,9]. We utilized transcriptomic cell types defined via the original SSv4 and 10X datasets for our evaluation, and the classification accuracy with binarized input data simulates the accuracy in a subsequent FISH experiment. In addition to the various gene selection methods, we also consider a panel of marker genes identified by Tasic et al.[46] for cell types in the

SSv4 dataset. For all gene selection approaches, larger panels enabled increasingly accurate cell type classification (Fig. 3). As expected, supervised methods that use cell type annotations during their selection procedure (e.g., MutInfo) perform better than unsupervised methods that use only unlabeled scRNA-seq data (e.g., Seurat). To emphasize this distinction between methods, we present results separately for unsupervised (Fig. 3A, C) and supervised methods (Fig. 3B, D).

Among the unsupervised approaches, PERSIST outperforms Seurat, Cell Ranger and GeneBasis for panels of all sizes. For example, when using 64 genes, PERSIST reaches 74% accuracy with the SSv4 data and 78% with the 10X data considering a total of 113 and 117 cell

A
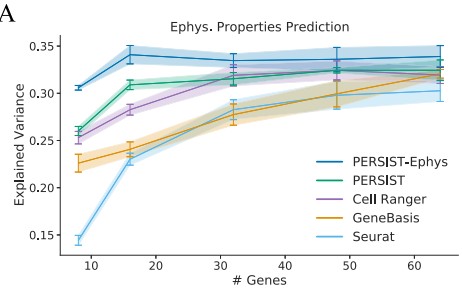
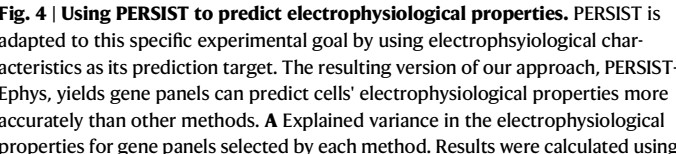

B
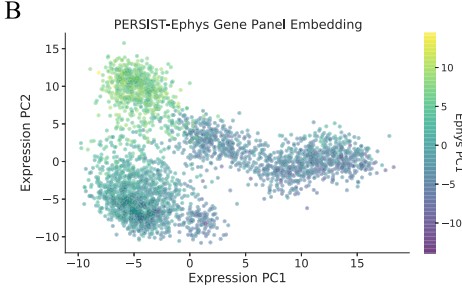

**Fig. 4 | Using PERSIST to predict electrophysiological properties.** PERSIST is adapted to this specific experimental goal by using electrophsyiological characteristics as its prediction target. The resulting version of our approach, PERSIST-Ephys, yields gene panels can predict cells' electrophysiological properties more accurately than other methods. **A** Explained variance in the electrophysiological properties for gene panels selected by each method. Results were calculated using a set of n=341 held-out cells, and error bars represent 95% confidence intervals determined by training with five bootstrapped datasets. **B** Low-dimensional embedding for the PERSIST-Ephys panel containing 64 genes, colorized according to the cells' electrophysiological properties (using the first principal component); the cells are effectively clustered according to their properties, with similar cells appearing nearby in the embedding space.

types, respectively. GeneBasis is the most competitive unsupervised baseline with the 10X data, and either GeneBasis or Seurat are most competitive with the SSv4 data. The gap in performance is largest for small panels, and the various methods roughly converge in accuracy for panels of 128 genes. As additional results, Supp. Figs. 7, 8 show confusion matrices that represent how cells of each type are most often classified, and Supp. Fig. 9 shows that distinct expression patterns for the selected genes are visible within each cell type.

PERSIST is an unsupervised selection algorithm by default, but we can also adapt it to cell type classification by using cell type labels as the prediction target during training (Fig. 1). This supervised version of our approach, termed PERSIST-Classification, matches or exceeds the performance of the other supervised approaches. For example, it reaches 81% accuracy in the SSv4 dataset and 82% accuracy in the 10X dataset using panels with 64 genes—a significant improvement over the unsupervised version. This illustrates the flexibility of our deep learning-based selection approach, and that PERSIST can be adapted to specific experimental objectives by simply adjusting its prediction target. The peak accuracy we observe with 128 genes is 85%, so our results also suggest that panels of these small sizes may be incapable of perfectly distinguishing fine-grained cell types, and therefore that FISH studies may benefit from analyzing more coarse clusterings.

Prior work suggests that the highly similar terminal nodes of the classification hierarchy may not all correspond to distinct cell types, but may instead reflect cell states or spatial gradients in gene expression[49]. We therefore divided cells into broader subclasses, which leads to greater classification accuracy because the groupings are more distinct and, trivially, there are fewer of them. For the SSv4 dataset, if we classify cells into 25 subclasses rather than the full 113 types, PERSIST-Classification reaches 84% accuracy using 16 genes (vs. 59% for 113 types) and 96% accuracy using 64 genes (vs. 81% for 113 types). In comparison, the unsupervised version of PERSIST provides accuracy just 3% and 1% worse, respectively (Fig. 3E–F). The results are similar but slightly less accurate when we classify into 50 subclasses. With a reduced number of cell type subclasses, Seurat, Cell Ranger, and GeneBasis are still not competitive with PERSIST, and PERSIST-Classification remains on par with the supervised procedures.

Although PERSIST does not match PERSIST-Classification in terms of classification accuracy, it is notable that PERSIST remains competitive despite not having access to cell-type labels. This indicates that PERSIST successfully captures cell-type information in an unsupervised manner. We attribute the strong cell type classification performance to our deep learning-based selection mechanism, which identifies non-redundant genes that help reconstruct the full expression profile, and to our use of gene expression binarization. In an ablation study, we also find that PERSIST's hurdle loss function is an important design choice, because it leads to better cell type

classification accuracy than training with mean squared error loss (Supp. Fig. 2). These results are promising because a consensus definition of cell types, and their continuous versus discrete nature, are far from settled[15,46,50]. Moreover, reference label information is currently available for only a handful of mouse and human tissues, and PERSIST can be used in an unsupervised manner in settings that lack established cell type hierarchies.

## PERSIST can be adapted to predict electrophysiological properties

As a further demonstration of our method's flexibility, we developed a specialized variant of PERSIST to identify marker genes that predict each cell's electrophysiological properties. For this purpose, we used a multi-modal Patch-seq dataset[14,51] containing transcriptomic and electrophysiological information from 3411 GABAergic neurons across 53 cell types in the mouse visual cortex. Specifically, the transcriptomic profile consists of the scRNA-seq counts for 1,252 curated genes, and the electrophysiological profile consists of a set of 44 sparse principal components (sPCs) summarizing different portions of the measurement protocol, as well as 24 biologically relevant features[51].

To select gene panels using the Patch-seq dataset, we used baseline methods that require only unlabeled expression data (Seurat, Cell Ranger, and GeneBasis) because the dataset lacks cell type annotations. For PERSIST, we first ran it in an unsupervised manner by selecting genes that can optimally reconstruct the full expression profile. Next, we also ran PERSIST in a supervised manner by using the vector of electrophysiological features as the prediction target, yielding a variant of our approach that we refer to as PERSIST-Ephys. We then investigated how well each gene panel represents a cell's electrophysiological profile.

As an evaluation metric, we attempted to predict the electrophysiological features of each neuron using the expression levels of the genes in each panel. Similar to previous experiments, we binarized gene expression levels to simulate applicability in a subsequent FISH study. We find that PERSIST-Ephys achieves the highest predictive accuracy, reaching a higher portion of explained variance with panels of all sizes (Fig. 4A). The unsupervised version of PERSIST is the second most accurate method, achieving comparable explained variance for larger panels, and Cell Ranger performs similarly in this case.

The strong performance of PERSIST-Ephys is primarily due to it being tailored to electrophysiological characterization. PERSIST-Ephys is designed to address this specific predictive task, but the other methods rely on selection criteria that are not explicitly related to electrophysiology. As a result, these methods picked distinct genes (Supp Fig. 12) and PERSIST-Ephys is roughly as accurate with a panel of 8 genes as the other methods are with 64 genes. None of the gene panels we tested exceed 35% explained variance, but notably, we find

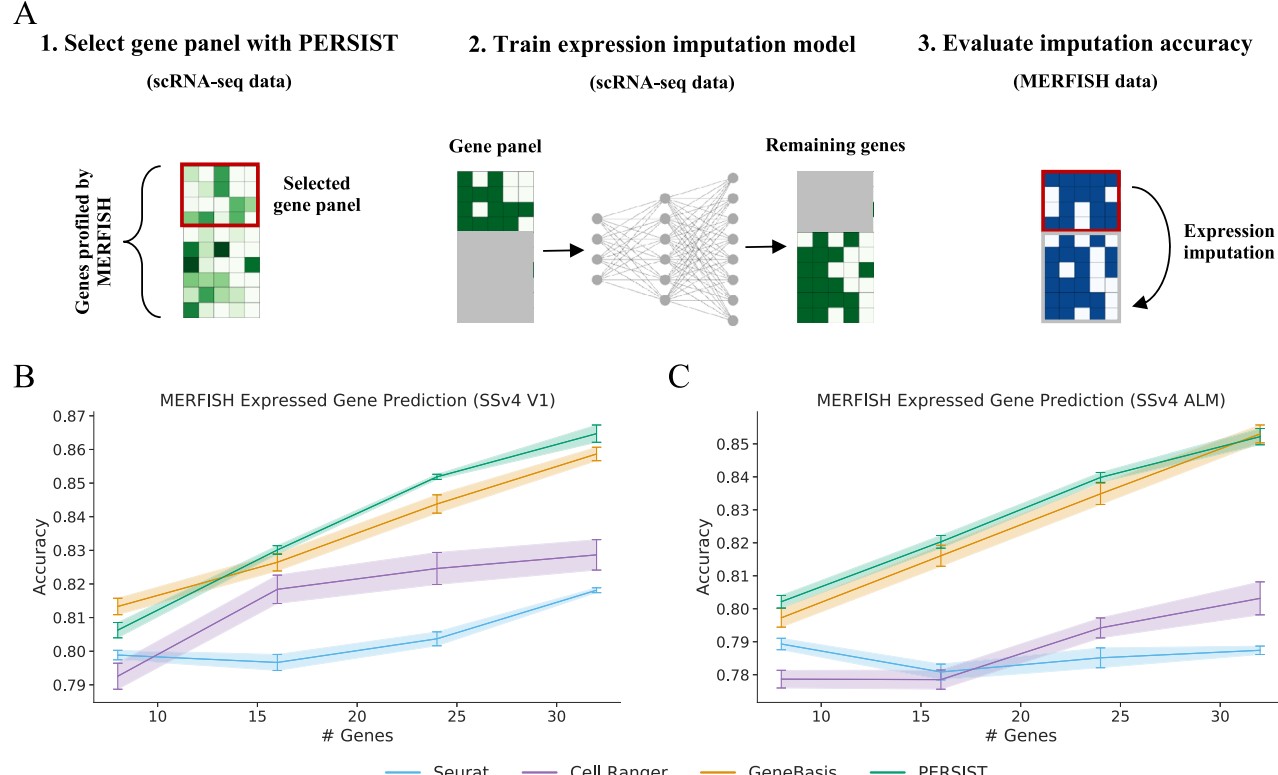

**Fig. 5 | Predicting MERFISH gene expression after training with scRNA-seq data.** To test whether models trained on scRNA-seq data can transfer to FISH despite the domain shift between technologies, we design an experiment combining SSv4 and MERFISH measurements. We find that binarizing gene expression levels enables accurate predictions in MERFISH cells, and that PERSIST outperforms prior gene selection approaches. **A** In this multi-step experiment, a gene panel is first selected using scRNA-seq restricted to the genes profiled in the MERFISH data. Next, an imputation model is trained to predict whether the remaining genes are expressed, again using scRNA-seq data. Finally, the imputation model's accuracy is tested on MERFISH data, using the small gene panel to predict the remaining genes. The imputation model's accuracy demonstrates that binarization enables successful transfer from scRNA-seq to FISH. **B** Expressed gene prediction accuracy for panels selected by each method when using the V1 SSv4 data. **C** Expressed gene prediction accuracy for panels selected by each method when using the ALM SSv4 data. The results were calculated using all cells from the MERFISH dataset (*n* = 280,327), and error bars represent 95% confidence intervals determined by training with five bootstrapped datasets.

that even the full set of 1252 genes does not exceed this level of accuracy; this suggests that the unexplained variance represents noise in the experimental results, or factors that are not captured by gene expression.

As an exploratory analysis, we also examined a low-dimensional embedding of the PERSIST-Ephys gene panel to understand the relationship between cells that are nearby in expression space. Our plot displays individual cells using the first two principal components of the panel containing 64 genes (Fig. 4B). To assess whether nearby cells have similar electrophysiological profiles, we colorized the cells by a scalar summary of the profile – the first principal component of the electrophysiological feature vector. The resulting plot reveals naturally occurring clusters of similar cells, which may be expected because PERSIST-Ephys selects genes whose expression is maximally indicative of the neuron's electrophysiological profile.

**Binarization enables gene expression prediction with MERFISH data**

PERSIST can identify informative marker genes for a variety of experimental objectives, but our previous evaluations used only scRNA-seq data due to the challenge of providing an unbiased comparison via FISH studies conducted with multiple panels. Nevertheless, such cross-modal experiments represent an essential use case, which is applying predictive models trained using scRNA-seq to data collected from spatial transcriptomics studies. Here, it is important to verify that binarizing expression levels enables such models to transfer successfully between technologies, which is difficult to ascertain because accompanying annotations are seldom available for FISH datasets (e.g., ground truth cell type labels, or expression levels of genes that are not part of the FISH panel). To investigate this question, we therefore devised a multi-step in silico experiment using the SSv4 scRNA-seq dataset in combination with data from a recent, large-scale MERFISH study[9].

In the MERFISH dataset, 258 genes were probed across 280,327 cells from the mouse primary motor cortex (MOp). Because ground truth cell type labels are not available, we instead chose to evaluate performance in the expressed gene prediction task (similar to Fig. 2D, F), where our goal is to predict which individual genes are detected in each cell. To do so, we first used the V1 and ALM SSv4 scRNA-seq datasets to select panels of 8-32 markers from within the Zhang et al.[9] MERFISH gene set. Then, an imputation model was trained—using only the scRNA-seq data—to predict which of the remaining genes are detected. Finally, using the resulting model, we predicted the set of detected genes in the MERFISH dataset (Fig. 5A). As in our previous experiments, we binarized both the scRNA-seq and MERFISH gene expression levels so the model trained with scRNA-seq data could transfer despite the measurement differences.

Encouragingly, we find that the scRNA-seq-trained models can predict expressed genes in the MERFISH data with high accuracy (Fig. 5B, C). The prediction accuracy tends to improve with larger panels; for example, PERSIST reached 86.5% with a 32-gene panel when trained on the V1 dataset. PERSIST in most cases achieves the highest

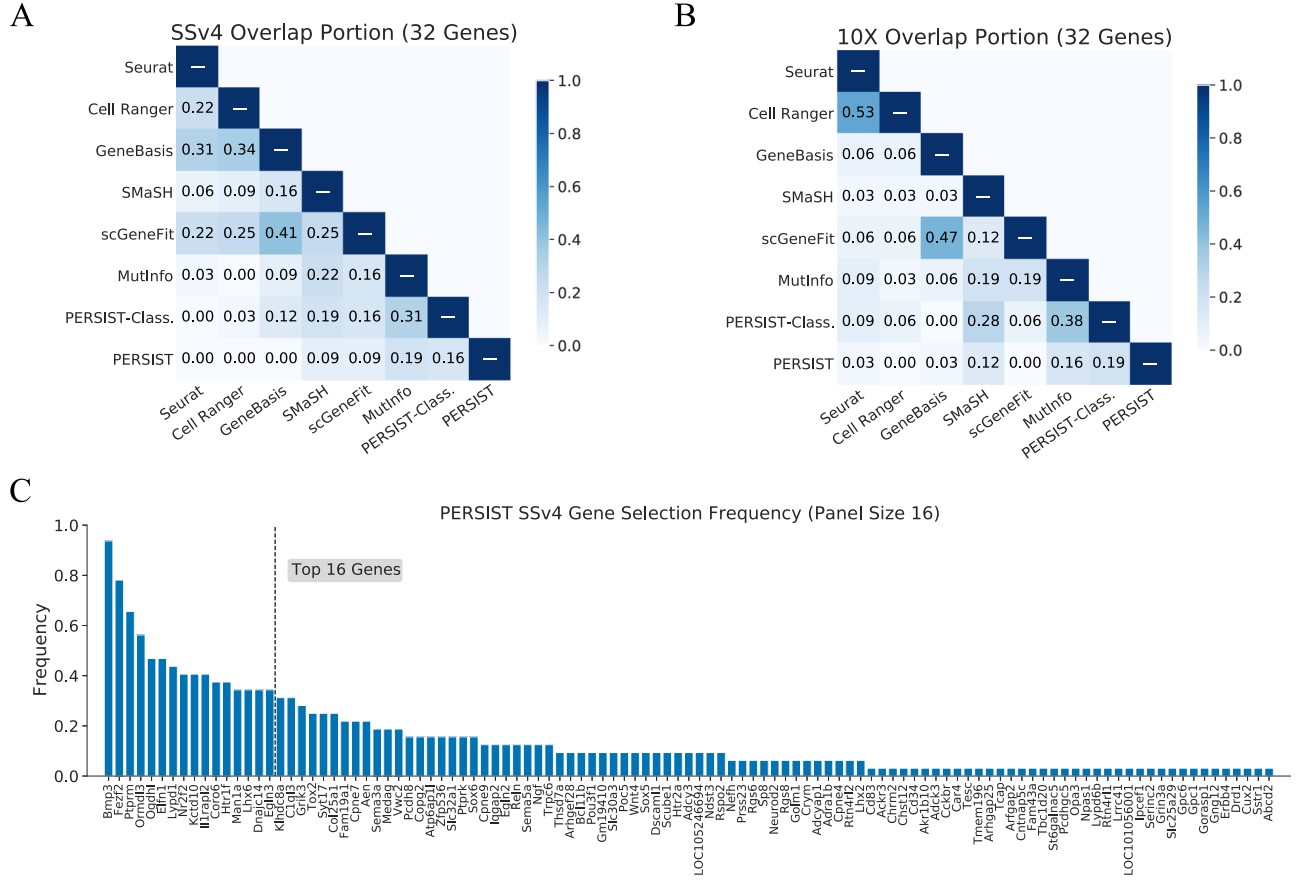

**Fig. 6 | Diversity in gene panels selected by each method.** Using the SSv4 and 10X datasets, we find that PERSIST and PERSIST-Classification select distinct gene panels from prior methods. Additionally, because PERSIST is non-deterministic across trials, we examine the frequency of its gene selections across multiple trials. **A** Portion of overlapping genes between panels of 32 genes for the SSv4 cells. **B** Portion of overlapping genes between panels of 32 genes for the 10X cells. **C** Frequency of gene selections within PERSIST panels containing 16 genes, aggregated across 32 independent trials.

accuracy, with GeneBasis being the most competitive baseline method. We also find that the prediction accuracy with the V1 cells is slightly higher than with the ALM cells. PERSIST not only outperforms other unsupervised methods that can operate with only unlabeled scRNA-seq data (Fig. 5), but also the approaches that leverage cell type labels when performing gene selection (Supp. Fig. 10). Crucially, the panels selected by all methods benefit from expression binarization to enable the imputation models to transfer across technologies.

In these experiments, we carefully determined the thresholds for binarizing the MERFISH expression counts by matching the quantile of the zero-threshold used with the scRNA-seq data. This procedure enables us to account for the zero-inflation present in scRNA-seq datasets. When we instead binarized the MERFISH data using a threshold value of zero, the prediction accuracy decreased for all gene panels (Supp. Fig. 10). Finally, because we have access to cells profiled using MERFISH with all genes in this case, we are able to determine that training the imputation model with MERFISH rather than scRNA-seq data results in an accuracy improvement of roughly 4–6% (see Supplementary Note 1). The gap is non-negligible, but it may be due in part to differences in the brain regions profiled in the reference dataset.

This evaluation approach does not represent a practical workflow for a real experiment, because the genes we predict are present in the panel; however, the fact that the model transferred successfully suggests that binarization can allow models for other predictive tasks to transfer from scRNA-seq to FISH data, including models for cell type classification or characterizing electrophysiological properties. This represents the first quantitative evidence, to our knowledge, that a predictive model trained exclusively with scRNA-seq

data can be transferred successfully to a subsequent spatial transcriptomics study.

## Variability in gene panel selections across algorithms

Because the gene selection algorithms tested here rely on diverse selection criteria, they produce different gene panels given the same reference scRNA-seq data. Here, we examine the overlap in gene panels selected by each algorithm, and we do so by calculating the proportion of overlapping genes within 32-gene panels chosen from among the 10,000 candidates in the SSv4 and 10X datasets. As expected, no two methods select the exact same gene set (Fig. 6A, B), but there is overlap among many pairs of methods. The probability of two random 32-gene panels sharing more than one gene is just $4.6 \times 10^{-3}$, so the overlap we observe suggests a shared reliance on a relatively small number of informative genes. The strongest similarity is between Seurat and Cell Ranger on the 10X dataset, at 53% overlap (17/32 genes); their overlap is lower with the SSv4 dataset, at just 22% (7/32 genes), and their similarity is perhaps due to the fact that both methods are based on per-gene variance levels. Another pair of similar methods is scGeneFit and GeneBasis, which have an overlap of 41% and 47% for the SSv4 and 10X datasets, respectively.

In comparison, we find that PERSIST has a relatively low overlap with other methods. The highest overlap for PERSIST is with PERSIST-Classification and MutInfo: PERSIST shares 19% of genes with PERSIST-Classification on the 10X dataset and 16% on the SSv4 dataset, and with MutInfo it shares 16% on the 10X dataset and 19% on the SSv4 dataset. Meanwhile, PERSIST-Classification and MutInfo achieve higher levels of overlap, at 31% for the SSv4 dataset and 38% for the 10X dataset;

similarly, PERSIST-Classification and SMaSH have 19% overlap on the SSv4 dataset and 28% overlap on the 10X dataset. For these three supervised methods, their similarity is likely due to their selection criteria that all aim to distinguish cell types. For panels containing either 16 or 128 genes, PERSIST's selections remain distinct from other methods, and they are still somewhat similar to those from PERSIST-Classification and MutInfo (Supp. Fig. 11). Overall, these results reflect that the various gene selection methods select distinct gene panels, and that PERSIST and PERSIST-Classification's improved performance across various metrics is enabled by the selection of substantially different panels.

Finally, we examined a somewhat unique characteristic of PERSIST: the stochasticity of its selections across runs. Because we implemented a deep learning model that is trained using stochastic gradient descent, the results from PERSIST and its supervised variants (PERSIST-Classification, PERSIST-Ephys) can differ across trials. This variability is somewhat unusual for a gene selection method, but this property is shared by other state-of-the-art feature selection techniques[20] and by the UMAP embedding method[52]. To examine the variability in the individual genes selected and the performance of the selected gene panels, we ran 32 independent trials of PERSIST with the SSv4 dataset.

For panels containing 16 genes, four genes were selected in at least half of the trials, with a single gene, *Bmp3*, being selected in 30 of 32 trials (Fig. 6C). The remaining ones were selected less consistently; the sixteenth most frequently selected gene was chosen in just over a third of trials, and 38 genes were selected just once. Reassuringly, differences in the composition between the gene panels had little impact on cell type classification accuracy or expression profile reconstruction (Supp. Fig. 13). The stability in performance despite changes in the gene panel composition is not surprising, because many genes have highly correlated expression patterns and thus can be readily substituted. In practice, we suggest running a small number of trials and then selecting the best trial using the validation loss achieved by the PERSIST deep learning model.

## Discussion

Identifying an effective gene panel is a pre-requisite for successful spatial transcriptomics studies. This work introduces PERSIST, which uses deep learning to select genes that are highly predictive either for the genome-wide expression profile or for a specific experimental objective. Our experiments with several datasets show that PERSIST selects more informative targets than existing algorithms, generally providing better predictive accuracy and/or enabling the use of fewer genes. In addition to our deep learning-based selection mechanism, a key contribution of this work is PERSIST's robust inference ability, which is achieved by using binarized gene expression levels: this helps mitigate the complex relationship between scRNA-seq and FISH measurements, and we find that it allows models to transfer to FISH studies despite the challenging domain shift. We also note that while our explicit demonstration is based on a recent MERFISH dataset, the problems addressed here are common to a broader class of spatial transcriptomic methods[8,10]. Therefore, our method is likely to improve target gene selection in a broader class of studies where genes are selected using surrogate data from a different technology.

From a computational perspective, PERSIST benefits from several aspects of deep learning that make it increasingly popular for data analysis in single-cell genomics[28,29,53]. These include the simplicity of gradient-based optimization, the scalability to large datasets enabled by minibatched training, and the flexibility of the prediction target and loss function. We profiled PERSIST's computational cost and found that both the running time and memory usage remain manageable when using 10,000 candidate genes and selecting a relatively large panel of 256 genes (Supp. Fig. 14). When using even larger datasets, PERSIST's computational cost can be managed by maintaining a smaller minibatch size, or by performing an initial filtering step to reduce the number of candidate genes. Finally, PERSIST does not require extensive tuning of parameters, and we used identical network architectures across all experiments (Supp. Table 1); however, future improvements may involve automatically setting all parameters to ensure maximum ease-of-use for both computational and biological users.

PERSIST is by default an unsupervised method that aims to reconstruct the genome-wide expression profile. Running it in an unsupervised fashion yields genes that are informative in general, but some information is necessarily sacrificed, because reconstructing the full expression profile using a small number of genes remains challenging. One interesting finding of our study is that while stochasticity in gene expression and detection limits the variance explained by the cell identity according to the reference clustering, relatively small PERSIST panels can explain more variance than the discrete cell type identity. This may suggest imperfections in the reference clustering, or biologically meaningful variability that lies in a continuum that is not captured by a discrete cell type label[54].

PERSIST can be tailored to arbitrary experimental goals by simply modifying the prediction target, thus enabling FISH studies to use gene expression as a bridge to other data modalities. Because alternative prediction tasks can be simpler than reconstructing many thousands of genes, using PERSIST in this fashion can enable the use of small gene panels that sacrifice minimal accuracy. Our experiments show examples with transcriptomic cell type classification and electrophysiological profile prediction, but other focused prediction tasks may include identifying disease properties or bridging with epigenetic information such as chromatin accessibility and methylation. Overall, PERSIST represents a powerful general-purpose solution for marker gene selection, and our design choices make it an effective tool for selecting small gene panels that can be used for FISH studies, and for spatial transcriptomics studies more generally.

## Methods

### Predictive and robust gene selection for spatial transcriptomics (PERSIST)

PERSIST aims to capture as much information as possible in a small gene panel, and it does so by selecting genes that can predict the genome-wide scRNA-seq expression profile. It relies on a deep learning model that reconstructs all the genes while using only a subset of the inputs, similar to an autoencoder but with a learned sparsity pattern. Deep learning-based reconstruction models have become popular for extracting low-dimensional embeddings in single-cell genomics[28,29,53], but PERSIST is designed to select a precise number of genes rather than fitting a general non-linear embedding. During the model's training, the input genes are selected by a custom network layer that enables training with stochastic gradient descent – a *binary mask layer*, which we describe below.

Deep learning is well suited to datasets with very large cell counts, and PERSIST's memory usage can be managed via the minibatch size used during training. In cases where memory usage is an issue, for example because the number of gene candidates and the intended panel size are both large, PERSIST can perform an initial filtering step using the *binary gates layer*, an alternative selection layer that we describe below. In addition to reducing memory usage, we find that such a two-stage approach provides minor performance improvements (Supp. Fig. 2), and we use this approach for the majority of our experiments.

### Hurdle loss function

Gene dropouts represent a significant source of noise in scRNA-seq data, and because PERSIST relies on scRNA-seq as a surrogate for FISH data, we must model gene dropouts appropriately when reconstructing the genome-wide expression profile. Recent computational tools

for scRNA-seq data have shifted away from using mean squared error loss[53,55] and towards zero-inflated models to account for this key source of noise[28,29,31,56,57]. In line with these works, we propose a loss function that can be applied to zero-inflated and continuous-valued measurements, which arise from common normalization approaches like as counts per million (CPM) normalization[58]. A zero-inflated negative binomial (ZINB) loss can instead be used if CPM normalization is not applied, but variability in the total UMI counts per cell can make integer transcript counts difficult to predict.

When reconstructing the genome-wide scRNA-seq expression profile, PERSIST trains a model that outputs predictions for each target gene $i = 1,...,d$. The predictions consist of a point prediction $\hat{y}_i$ as well as a probability $\hat{p}_i$ of the gene having non-zero expression. Given a fixed weighting parameter $\gamma > 0$, the hurdle loss $\ell_\gamma^{(i)}$ for gene $i$ with observed expression level $y_i$ is defined as

$$\ell_\gamma^{(i)}(\hat{p}_i, \hat{y}_i, y_i) = \begin{cases} \frac{1}{\gamma}(\hat{y}_i - y_i)^2 - \log \hat{p}_i & y_i > 0 \\ -\log(1 - \hat{p}_i) & y_i = 0. \end{cases} \quad (1)$$

The loss has a cross-entropy component for predicting whether the gene is expressed or not, as well as a mean squared error component that is incorporated only when the gene is expressed ($y_i > 0$). Because we use log-normalized gene counts $y_i$, the loss $\ell_\gamma^{(i)}$ can be understood as the negative log-likelihood for a log-normal hurdle distribution[30,59], where we implicitly simplify the log-normal component by assuming fixed standard deviations for each gene, and the weighting parameter $\gamma > 0$ controls the trade-off between predicting whether a gene is expressed and predicting its expression level. The total loss $\ell_\gamma$ for the full expression profile is the following, where $\hat{y}, \hat{p}$ and $y$ represent vectors of predictions and expression levels for all genes:

$$\ell_\gamma(\hat{p}, \hat{y}, y) = \sum_{i=1}^m \ell_\gamma^{(i)}(\hat{p}_i, \hat{y}_i, y_i). \quad (2)$$

In practice, we fix $\gamma = 10$ so the mean squared error and cross-entropy components of the loss have similar scale, but this parameter can also be tuned.

## Feature selection layers

We introduce the *binary mask layer* as a tool to select a user-specified number of inputs within a deep learning model. Our approach is based on the Concrete distribution[34] (also known as the Gumbel-Softmax[35]), which lets us optimize discrete probability distributions using stochastic gradient descent. To select exactly $k$ genes from $d$ candidates, we multiply the model input by a binary mask generated using the element-wise maximum of $k$ Concrete random variables, which are denoted as $A_i \sim \text{Concrete}(\alpha_i, \tau)$ for $i = 1,...,k$ (Supp. Fig. 1A). Each Concrete distribution is parameterized by unnormalized probabilities $\alpha_i \in \mathbb{R}_+^d$ and a temperature value $\tau > 0$, and each one converges to a multinomial distribution with probabilities given by $\frac{\alpha_i}{\mathbf{1}^\top \alpha_i}$ as $\tau \to 0$[34].

The binary mask layer's input is a vector of binarized gene expression levels $x \in \mathbb{R}^d$, and its output is given by the element-wise product $a \odot x$, where $a = \max 2.2214 pt a_i \in \mathbb{R}^d$ is the element-wise max of samples $a_i$ from each Concrete random variable. The layer is followed by a neural network $f_\theta$ that predicts the label $y$ given $a \odot x$, and we train the model by optimizing the following objective:

$$\min_{\theta, \alpha} \mathbb{E}_{x,y,a} \left[ \ell_\gamma(f_\theta(a \odot x), y) \right]. \quad (3)$$

To select a specific number of genes, we need only ensure that the temperature parameter $\tau$ is sufficiently low at the end of training. In practice, we find that each Concrete random variable concentrates its probability mass on a single input, yielding a set of exactly $k$-selected

genes. The binary mask layer is similar to the CAE approach[20], but we find that our parameterization, which uses element-wise multiplication rather than a matrix multiplication, provides slightly better results (Supp. Fig. 2).

We also introduce the *binary gates layer* as a memory-efficient alternative to the binary mask layer. Similar to previous work[36], we learn a Binary Concrete random variable for each input feature, denoted as $B_i \sim \text{BinConcrete}(\beta_i, \tau)$, and we use these gate variables to perform element-wise multiplication with the corresponding genes (Supp. Fig. 1B). In the Binary Concrete distribution, each parameter $\beta_i > 0$ represents an unnormalized probability, $\tau > 0$ represents a temperature value, and each random variable $B_i$ converges to a Bernoulli random variable with probability parameter $\frac{\beta_i}{1 + \beta_i}$ as $\tau \to 0$[34].

The binary gates layer output is given by the element-wise product $b \odot x$, where $x$ is the input and $b = [b_1, ..., b_d] \in \mathbb{R}^d$ is a vector of samples $b_i$ from each random variable $B_i \sim \text{BinConcrete}(\beta_i, \tau)$. The layer is followed by a neural network $f_\theta$ that predicts the label $y$ given $b \odot x$. Genes are eliminated when we learn low $\beta_i$ values, and we encourage this by augmenting the loss function with a penalty on the BinConcrete samples:

$$\min_{\theta, \beta} \mathbb{E}_{x,y,b} \left[ \ell_\gamma(f_\theta(b \odot x), y) + \lambda \mathbf{1}^\top b \right]. \quad (4)$$

The regularization term penalizes the number of selected genes, and the hyperparameter $\lambda > 0$ controls the trade-off between prediction accuracy and the number of genes used. Determining a specific number of genes requires choosing the correct $\lambda$ value, and our implementation finds this value automatically by iteratively adjusting $\lambda$ using the secant algorithm[60]. Briefly, given a desired number of candidate genes $d' < d$, our method iteratively updates the $\lambda$ value based on the number of genes yielded by previous $\lambda$ values. In our experiments with the SSv4 and 10X datasets, we initially narrow the set of candidates to roughly 500 genes. For the experiments involving Patch-seq and MERFISH data, which involve fewer candidate genes, we directly select gene panels using the binary mask layer.

## Training

When training our deep learning models, we perform optimization using Adam with the standard learning rate ($10^{-3}$)[61]. Over the course of training, the temperature parameter $\tau$ is geometrically annealed from a high value to a low value to encourage discrete feature selection, similar to previous work[20]. For both input layers, we use an initial temperature $\tau = 10.0$ and a final temperature of $\tau = 0.01$. After training, the parameters for the Concrete random variables are naturally learned such that the same genes are selected at every forward pass. That is, most Binary Concrete variables are deterministically equal to 0 or 1, and the Concrete variables are one-hot at the same entry in every sample. As a selection criterion for the binary gates layer, we retain all genes $i$ such that $\frac{\beta_i}{1 + \beta_i} > \frac{1}{2}$. For the binary mask layer, we select $k$ genes using the maximum index of each vector of unnormalized probabilities $\alpha_i$ for $i = 1,...,k$.

The binary mask layer is necessary for selecting a specific number of genes, but the binary gates layer is preferable for eliminating a large number of genes. The binary mask layer has $k \times d$ learnable parameters when selecting $k$ genes, whereas the binary gates layer has only $d$. The binary mask layer can therefore be difficult to apply directly in scenarios where $d$ and $k$ are both large, and the binary gates layer is useful for our datasets with $d = 10,000$ total candidate genes (SSv4 and 10X). In practice, the outcome from running PERSIST may differ across runs when the random seed is not fixed. For our experiments, we run five trials and use the gene panel that achieves the best validation loss.

## Ablation experiments

We tested several variants of PERSIST to validate our design choices. First, we compared PERSIST to a version that skips the initial step of narrowing the set of candidate genes and proceeds directly to training with the binary mask layer. The results are slightly worse (Supp. Fig. 2), which we attribute to the beneficial effect of iteratively reducing the number of genes. Next, we compared PERSIST to the CAE approach[20], which is closely related but differs in its choice of loss function (mean squared error) and feature selection layer. We find that the CAE underperforms PERSIST across our three evaluation metrics. Most noticeably, the CAE underperforms at cell type classification (Supp. Fig. 2C), which is consistent with recent work that highlights the importance of modeling gene dropouts[28,29]. Finally, we observe that binarizing gene expression counts improves the CAE's performance on our metrics, but a small gap remains for expressed gene prediction, that the binarized CAE still does not match PERSIST's performance for cell type classification.

## Memory and running time

To benchmark PERSIST's computational cost, we ran it across a variety of parameter settings and measured the running time and GPU memory usage. For simplicity, we used the SSv4 training set with 17,728 cells, we fixed the minibatch size to 128, and we selected panels by training directly with the binary mask layer for 500 epochs. Our models were all trained on a single NVIDIA GeForce RTX 2080 Ti. Supp. Fig. 14 shows the results from varying two key parameters related to the dataset size. First, after fixing the panel size to 32, we varied the number of candidate genes from 2000 to 10,000. The results show that the memory scales linearly, and the run-time scales sublinearly in the number of candidates. Next, after fixing the number of candidate genes to 10,000, we varied the panel size from 32 to 256. The results show that the memory scales linearly, and that the run-time scales roughly linearly. We did not test different numbers of cells because this does not affect memory usage, and the run-time would simply depend on how the number of epochs is tuned to each dataset size.

Across all the settings tested, PERSIST's run-time is not prohibitive. The GPU memory usage can become high when both the number of candidates and the panel size are large (e.g., 10,000 candidates and panel size of 256), but memory issues can be mitigated by either reducing the minibatch size or performing an initial filtering step using the binary gates layer. Compared to other gene selection methods, certain methods are faster than PERSIST because they do not require training a model (Seurat, Cell Ranger) or involve a simpler model that does not perform gene selection (SMaSH), but other methods can be slower, particularly with large datasets, because they involve expensive greedy heuristics (MutInfo, GeneBasis).

## Expert knowledge and supervision

PERSIST can be used in a purely data-driven fashion, or it can be used while incorporating expert knowledge. For example, the candidate genes can be restricted to those with known biological function or other desirable properties, or the gene panel can be forced to include certain hand-selected genes. When such expert knowledge is applied to the set of candidate genes, PERSIST finds the best available panel among the current candidates and while accounting for any pre-selected genes.

For example, when working with specimens of lower RNA quality (e.g., post-mortem human samples), it can be useful to consider only highly expressed genes that can be readily detected. Supp. Fig. 15 shows that when PERSIST is restricted to using only genes whose maximum expression level is higher than the median value, we achieve roughly the same performance as when choosing from among all 10,000 candidate genes in the SSv4 dataset. In general, filtering steps that preserve a large number of gene candidates should not prevent PERSIST from finding highly informative panels.

Rather than using PERSIST in an unsupervised manner, we can also incorporate supervision to align the gene panel with specific experimental aims. If the spatial transcriptomics study has a specific prediction objective, such as characterizing electrophysiological properties, and accompanying labels are available for the reference scRNA-seq data, PERSIST can incorporate this information into the selection procedure. Adapting PERSIST is straightforward, requiring only a change in the prediction target and loss function, as we demonstrate with PERSIST-Classification (multiclass cross entropy loss, see Fig. 3) and PERSIST-Ephys (mean squared error loss, see Fig. 4).

## Models and hyperparameters

Each dataset is split into training, validation and test sets. The training and validation sets are used to select gene panels and train predictive models, and the test set is used only to evaluate the performance of trained models. Our hyperparameter choices were all made using the validation set, including for early stopping. For PERSIST and its supervised variants, we used identical neural network architectures across all datasets, and the only hyperparameters we adjusted are the number of training epochs and the minibatch size (Supp. Table 1). Our model choices for the various downstream tasks are shown in Supp. Table 2, where we use multi-layer perceptron (MLP) models for most tasks, and LightGBM models[62] for cell type classification. Confidence intervals for the downstream tasks were determined by training models with five bootstrapped training sets and measuring the test set performance across these models.

When using PERSIST, overfitting can be avoided by using network architectures that are not too large and performing early stopping during training. If overfitting is a significant concern, additional regularization techniques are straightforward to incorporate, including dropout, batch norm and L1/L2 regularization. Finally, when fitting models for the downstream predictive tasks, such as cell type classification, one can further mitigate overfitting by using a non-neural network model with fewer learnable parameters[63].

## Evaluation metrics

The variance explained by a gene panel is measured by training a neural network to predict the full expression profile, measuring the mean squared error on the test set, and subtracting this quantity from the total variance. We provide results on an intuitive scale by calculating explained variance as a portion of total variance. For this metric, we used CPM normalized and logarithmized expression counts as the prediction target. The portion of explained variance does not approach 100% for any method, but this is in large part due to the stochasticity of gene expression and measurement; as described in the main text, the ground truth cell type labels explain only 19% and 11% of the variance for the SSv4 and 10X datasets, respectively.

The cell type classification performance is the accuracy of a gradient-boosted decision tree (GBDT) model trained with a multiclass cross-entropy loss. The cell type with the highest classification probability is taken as the predicted class. To evaluate performance with cell subclasses for the SSv4 dataset, we trained models with cell types merged according to their order in the transcriptomic hierarchy (Supp. Fig. 16).

Expressed gene prediction accuracy is measured using a neural network trained to separately predict whether each gene is expressed. Similar to a classifier model, the loss function is a per-gene cross-entropy loss that is then added across genes. A gene is predicted to be expressed if the network's probability exceeds 0.5, and we calculate the accuracy by calculating how often the predictions agree with the true expression, and averaging the results across all target genes.

## Data pre-processing

Due to technical noise in scRNA-seq data, we applied CPM normalization[58] to the raw measurements and then applied the log1p operation. These values serve as prediction targets for PERSIST, whereas the inputs to our various models are binarized expression counts. For both the SSv4 and 10X datasets, we restricted our analysis to the 10,000 transcripts with the highest variance. We used exon counts for the high-resolution SSv4 dataset and the sum of intron and exon counts for the lower-resolution 10X data. For the MERFISH dataset, the only pre-processing we applied was gene expression binarization. When working with the MERFISH data, we analyze 253 genes that appear in both the Zhang et al.[9] and SSv4 datasets rather than the 258 genes described in the original work.

## Expression quantization

We quantized gene expression levels in order to train models using scRNA-seq data that can transfer to FISH studies. There is a complex domain shift between the two technologies, but their quantized gene expression values should be similar if we assume that the transformation between measurements is monotonic. When applying models trained with scRNA-seq on FISH data in practice, we recommend using a threshold matching approach, i.e., finding the quantile that the scRNA-seq threshold represents in the scRNA-seq measurements (we use a threshold value of zero), and then identifying the matching threshold in the FISH data. This approach is used for the results in Fig. 5, and its utility is demonstrated in Supp. Fig. 10.

## Baseline methods

For the Seurat and Cell Ranger gene selection protocols, we used the implementations available in the Scanpy[40] package: https://github.com/theislab/scanpy. Seurat was run with raw gene expression counts and Cell Ranger with logarithmized counts. For the scGeneFit method, we used the authors' implementation with logarithmized expression counts, and we used the default hyperparameters: https://github.com/solevillar/scGeneFit-python. For the GeneBasis method, we used the authors' R implementation: https://github.com/MarioniLab/geneBasisR. For the SMaSH method, we used the authors' Python implementation with a neural network architecture identical to PERSIST, and with feature importance scores calculated using DeepSHAP[64]: https://gitlab.com/cvejic-group/smash. Finally, for the MutInfo method, we implemented the greedy forward selection algorithm described in prior work[42,65] using the hyperparameter $\beta = 1$ to account for gene correlations.

For the panel of marker genes used in Fig. 3B, we used a set of genes identified in various tables of Tasic et al.[46]. The original work listed 77 such markers, and we used the 59 that were represented in our dataset after narrowing it to 10,000 high-variance genes.

## Statistics & reproducibility

The samples from each data source were assigned at random to training, validation, and test splits, and to obtain unbiased statistics we used the test data only when calculating evaluation metrics. No data was excluded from the datasets. Uncertainty estimates were provided throughout the experiments by fitting models with boostrapped training sets. Code for running our method is available online (see "Code availability"), and the various datasets are available to download online (see "Data availability").

## Reporting summary

Further information on research design is available in the Nature Portfolio Reporting Summary linked to this article.

## Data availability

The datasets used in this work are summarized in Supp. Table 3, including the species, brain regions, and annotations used for our experiments. The V1/ALM SmartSeq mouse neocortex data is available at https://portal.brain-map.org/atlases-and-data/rnaseq/mouse-v1-and-alm-smart-seq. The M1 10X data is available at https://portal.brain-map.org/atlases-and-data/rnaseq/human-m1-10x. The Patch-seq data is available at https://github.com/AllenInstitute/coupledAE-patchseq. The MOp MERFISH data is available at https://download.brainimagelibrary.org/02/26/02265ddb0dae51de/. Source data are provided with this paper.

## Code availability

Source code for PERSIST is provided at https://github.com/iancovert/persist/, along with tutorial notebooks and examples of data pre-processing code. The repository contains a list of software dependencies, and PERSIST is implemented in PyTorch (version 1.13.1). The code is also archived at https://doi.org/10.5281/zenodo.7714685.

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

## Acknowledgements

We thank members of the Lee Lab, Jennie Close, and Polina Kosillo for helpful feedback on the manuscript. This work was funded by NSF DBI-1552309 and DBI-1759487 (I.C. and S.-I.L.), NIH R35-GM-128638 and R01-NIA-AG-061132 (I.C. and S.-I.L.), NIH 1RF1MH128841-01 (K.S. and U.S.), the Allen Institute (R.G., K.S., and U.S.), the Howard Hughes Medical Institute (T.W. and K.S.) and the Helen Hay Whitney Foundation (T.W.). R.G., K.S.,

and U.S. wish to thank the Allen Institute founders, Paul G. Allen, and Jody Allen, for their vision, encouragement, and support.

## Author contributions

I.C., U.S., and S.-I.L. designed the primary algorithm. I.C., R.G., U.S., and S.-I.L. designed the experiments, and I.C. and R.G. carried out the analyses. I.C., R.G., T.W., K.S., U.S., and S.-I.L. wrote and edited the manuscript. K.S., U.S., and S.-I.L. supervised the study.

## Competing interests

The authors declare no competing interests.
