## [Peer Review File · Nature Communications]

Reviewers' Comments:

Reviewer #1:

Remarks to the Author:

In "Predictive and robust gene selection for spatial transcriptomics", Covert and colleagues present a framework for gene panel selection that leverages single-cell sequencing datasets. Their machine learning approach employs an autoencoder with a mask layer to find gene panels that best predict the genome-wide expression profiles of single cells. The authors test this approach on two neuron-focused single-cell datasets and use the panels to classify cell types and estimate electrophysiological data. While some of the methodological ideas in this paper are interesting, the validation or performance estimate are unusually weak.

Major issues:

1) The authors do not test the autoencoder in external datasets and cannot gauge whether it is overfitting or overtraining. Recently, Grabsky and Irazarry highlighted the pitfalls of within dataset learning of cell type annotations by comparing performance on external and internal withheld test cells (<https://doi.org/10.1093/biostatistics/kxac021>). They examined the effects of noise (incorrect labels), downsampled counts, and cells. At a minimum, the authors should similarly investigate the effects of within dataset evaluations, undertake cross-dataset evaluations, and significantly revise the manuscript accordingly.

2) Evaluation of the other methods using the loss function created for the PERSIST framework is somewhat circular. Unlike the PERSIST model, these approaches were not explicitly designed to identify panels that explain the variance of the top 10,000 most variable genes using binarized expression. In essence, there is a question about the generalizability of the performance statistic (particularly when one model is optimized with respect to it, even if that is changed to be evaluated across datasets).

3) The strong separation in the results and narrative between supervised and unsupervised approaches seems unnecessary to me, given that the goal is to identify gene panels and not annotate cell types. It is an expected result that cell-type annotations will help, if available. If they are not available, it likely makes sense to focus on cell-type discovery first. Alternatively, it would make sense to develop a better ensemble of use cases that plausibly map to unsupervised panel selection. In many ways this manuscript felt like a white paper on panel selection for a particular high-throughput experiment. A broader zone of application would be helpful.

4) Although the authors examine different levels of granularity, the results do not provide information about performance across cell types with varying proportions. This could be achieved by providing confusion matrices, micro, and macro averaged metrics.

5) The authors mention that PERSIST requires a significant amount of memory and that it is difficult to run on a GPU. Deep learning models are well suited for GPU's and PERSIST would not be considered 'deep'. It is also odd that the demo notebook is set to use only 128mb of GPU memory. Additionally, the binarized input expression data should be relatively small. Given that the number of genes that can be spatially assayed is expected to increase, details on these computational limitations should be provided.

I question the usability of PERSIST given the computational limitations and the lack of integration with single-cell software libraries. In addition, as Table 1 shows, a significant amount of tuning was undertaken to customize PERSIST to various tasks. I attempted to get the provided demo notebook running on my macbook pro but gave up after failing to load the dependencies. The average lab seeking to identify a panel will not have the expertise to apply PERSIST in its current state. For example, the demo notebook is full of hardcoded variables that require machine learning expertise to understand ('hidden=[128, 128]' and 'max_nepochs=500'). Creating a well-documented Google Colab notebook might help in this regard.

6) Although not ground truth, the cell-type labels from the MERFISH dataset should be used for both training and evaluation.

7) The Discussion section, which currently consists of two paragraphs, should be expanded to mention scalability, likely sources of error, and future directions. In addition, the generalizability of PERSIST beyond neurons should be discussed as the application to other datasets with cell types with less expressed genes may limit accuracy.

In general, the novel approach taken by the authors for gene set selection may be of interest from others searching for optimal gene panels. However, a better case needs to be made given the limited accessibility and marginal performance gains in supervised settings.

Reviewer #2:

Remarks to the Author:

Covert et al describe in this manuscript a novel approach to feature selection from single-cell data, primarily single-cell RNA sequencing (scRNA-seq), which they term PERSIST. In brief, PERSIST is a machine learning approach that aims to identify a user-defined number of genes that best capture the diversity of expression observed in scRNA-seq data or, in alternative versions, aims to capture the features that themselves best capture companion labels or measurements, such as the labels applied after cell-type identification. The authors place this algorithm in the broader context of the growing single-cell field by highlighting the role it could play in guiding the design of targeted transcriptomics measurements such as the modestly or highly multiplexed FISH assays, e.g. MERFISH, seqFISH, osmFISH.

The authors well describe the development of their algorithm and then benchmark it against other approaches for identifying informative gene sets from scRNA-seq, namely the methods for selecting highly variable genes provided in the Seurat and Cell Ranger packages as well as the GeneBasis algorithm. In an interesting aside, they very nicely show that the PERSIST framework can be used to capture genes that are most associated with electrophysiological properties of single cells measured in Patch-seq. Finally, they explore the ability of PERSIST to identify genes that can be leveraged to train imputation models to predict the expression of MERFISH measurements in the mouse cortex based on companion scRNA-seq measurements in the same region.

Overall, given the now wide and standard use of scRNA-seq and the still unmet need in those measurements to identify optimal and most-informative gene subsets in combination with the growing excitement and use of targeted image-based approaches to single-cell transcriptomics, I believe that the PERSIST approach provided here will be of interest to a range of scientists, and I support the publication of this manuscript.

However, I have several comments and concerns that I believe that the authors will want to consider in a revision of their manuscript. I list them below organized into major and minor concerns.

Major concerns:

1) The authors clearly motivate the PERSIST algorithm for the purpose of leveraging scRNA-seq to select features for multiplexed FISH assays. I completely agree that this application is an important one. Yet, most of their manuscript does not deal with FISH data, and, in the context, of the FISH data they do explore, they leverage PERSIST to identify a minimal set of genes that can be used to effectively co-embed scRNA-seq with MERFISH data in order to predict gene expression. Unfortunately, in no context do the authors test the ability of their algorithm to guide the design of FISH gene panels in a direct fashion. I do not fault the authors for this point, as it is likely not possible to use their method to design panels that are then experimentally tested as the FISH-based assays are not yet widely available. However, perhaps the manuscript would be stronger and easier for the average reader to understand if the authors could place greater emphasis on the other applications in which they validate PERSIST?

2) The authors benchmark the feature selection process of PERSIST against other methods for selecting informative gene sets, and this benchmarking is quite thorough. However, I have some concerns with the way in which they discuss the comparison between their algorithm and the

methods used for selecting genes with Seurat and CellRanger. Critically, the goal of the feature selection algorithms in Seurat and CellRanger seems to me to be very different than that of PERSIST. These approaches aim to reduce the number of genes in subsequent calculations in order to reduce noise and the amount of computation. As such, these approaches are not looking to remove redundancy in the information carried by the genes within their final 'highly variable' set and, in many ways, they do not aim to extensively minimize the number of genes. By contrast, I suspect that implicit to the PERSIST approach is the removal of genes from the final set that carry redundant information as this ability is what would allow small targeted panels to carry information on the expression of the entire transcriptome. Thus, I suspect that—almost by design—CellRanger and Seurat should perform very poorly for small numbers of genes and using them as a benchmark in this context might not be as appropriate. I think that the authors should provide a more nuanced interpretation of what this comparison between PERSIST and Seurat/CellRanger implies about the performance benefits of their method.

3) Along these lines, identifying the most variable genes has not been the way in which others have leveraged scRNA-seq to guide the selection of markers for image-based spatial transcriptomics panels. A more common approach is to first cluster the scRNA-seq data, then to derive marker genes for each cluster, i.e. genes that are differentially expressed within that cluster, relative to all cells not within that cluster. Of course, this approach discards genes that capture inter-cluster variability. Nonetheless, given the central importance the authors have placed on the possibility of using PERSIST to identify the most informative genes for spatial transcriptomics panels, I think it is important to have at least a minor exploration of the overlap between the genes produced by PERSIST and marker genes derived from the scRNA-seq data in this fashion.

4) Given the central role of scalability in the usefulness of many computational methods, it would be very helpful to the average reader to have some quantification of the computational time and memory usage for the different examples explored in this manuscript. It is also important to have some comment on how resource use will scale as the number of genes/cells increases.

5) I had a difficult time understanding how to interpret the performance of PERSIST (and the other methods) in terms of the explained variance (Fig 2A,C,E), as the fraction of the explained variance is low in all cases. On one hand, this observation may not be surprising as I would anticipate that a sizeable fraction of the observed variance is due to the stochasticity of gene expression and detection. Yet, given the stated goal of PERSIST, could the authors provide some comment on how much variance they anticipate is meaningful biological variance?

6) I also had a difficult time understanding the metric they provide for the accuracy of their predictions. My central concern here is two-fold. First, the definition of this calculation in the methods portion of the manuscript is concise and technical and would benefit substantially from more explanation. Second, it strikes me that a more straightforward calculation of accuracy is both possible and, perhaps, more meaningful. Specifically, for each gene, could the authors calculate what fraction of the cells expressed that gene when they were predicted to express it, what fraction express that gene when they were predicted not to, and what fraction of cells don't express that gene when they were predicted to? Do these values depend on properties of the gene, such as its average expression? Overall, given the low capture efficiency and digital nature of scRNA-seq data (i.e. a high drop out rate), I am surprised that accuracy values can be so high. To illustrate my point, one could easily imagine that over 70% of the cells within a given cluster in single-cell RNA sequencing might have no copies of an RNA even when that gene is expressed on average at meaningful levels within that cluster. Again, given the stochastic nature of expression, I would anticipate that it is hard to make predictions of the expression of this gene that are correct 70% of the time or more. Arguments like this one motivate my suggestion that they authors explore more intuitive measures of correspondence between the measured and predicted values could be useful.

More minor concerns:

1) I take some issue with the description of the state-of-the-art for multiplexed FISH assays. In particular, the authors highlight that even in the largest surveys these methods target less than 5% of the transcriptome and in more typical studies less than 0.2%. Yet, both MERFISH and seqFISH+ have published measurements of ~10,000 transcripts, which is far higher than 5% of the transcriptome. Moreover, relatively standard measurements now target several hundred genes if not ~1000, which is more than 0.2% of the transcriptome.

2) The numbering and labeling of the supplemental figures was difficult to follow and should be

fixed. For example, supplemental figures are not labeled as such in the captions but only in reference in the text, and these figures are called out of order.

3) On page 10, the authors state that “this represents the first evidence, to our knowledge, that a predictive model trained using only scRNA-seq data can be applied successfully in a spatial transcriptomics study”. I strongly suggest the authors consider softening this statement because there are many methods that have demonstrated co-embedding of scRNA-seq and spatial transcriptomics data and that such co-embedding can be used for information transfer between these different data sets. Thus, the notion of combining these two types of data modalities via a variety of computational methods is not new.

4) The authors state on page 17 that “it can be useful to consider only genes with high expression levels, because these transcripts are easier to identify in a FISH experiment.” However, for the vast majority of FISH techniques, especially the highly multiplexed methods, the opposite is true. It is typically far harder to address very highly expressed genes than lowly expressed genes.

Reviewer #3:

Remarks to the Author:

The authors introduce PERSIST (predictive and robust gene selection for spatial transcriptomics), a DL-based feature selection method aiming to inform targeted spatial transcriptomics (ST). This approach has tremendous potential in facilitating targeted ST protocols, such as FISH, given that a set of genes must be identified a priori.

Strengths:

- I believe the idea of binarizing the scRNAseq to account for the binary nature of FISH data and using a Gumble-Softmax distribution (initially used for developing categorical/binary VAEs) to optimize the model given the binary inputs is novel.

- Most non-linear dimensionality reduction approaches for scRNAseq (e.g. DCA) suffer from unexplainable latent spaces, which could result in limited utility based on the use case. I believe having dimensionality reduction defined as the number of genes that explain the most variance is appropriate for this work and the desired application.

- The results on scRNAseq reconstruction and cell-type classification based on gene panels are very interesting and significant. However, I do believe more appropriate comparisons are missing (more on this in the Major Comments/Questions Section).

- I found gene expression imputation with MERFISH data fascinating. Though I believe the feature selection comparisons are incomplete (detailed in the Major Comments/Questions Section), this experiment shows the potential of transferring information from scRNAseq to other modalities, which can be tremendous.

Major Comments

1) My main concern is the lack of comparison with existing supervised/unsupervised gene selection methods for scRNAseq, which have shown to accurately narrow down the gene space based on a desired downstream task. For example, some popular single-cell methods such as scVI [Lopez et. al] can be modified to narrow down the gene space based on the importance of the genes to a downstream task (in the case of scVI, this can be done if one uses a linearly-decoded VAE (LDVAE)). Another recent DL-based approach, called N-ACT (Heydari et al.), uses neural attention to identify the most important genes in optimizing an objective. It seems that the same framework can be leveraged to optimize the same loss function used in this work to reconstruct the scRNAseq data, and after training, the n most important genes can be selected and used for targeted ST.

The authors do not compare their methods to the mentioned approaches, nor do they provide justification of why such comparison would not be appropriate. I believe adding such comparisons/justifications would distinguish this work from the existing approaches and further strengthen the manuscript. I would appreciate learning more about the authors' response to this comment before further evaluating the feature selection results.

2) Could the authors justify why (or if) it is appropriate to use binarized data as inputs to Seurat and Cell Ranger? I understand the authors' desire for removing data processing as a possible explanation for PERSIST's better performance, but I am not convinced that using bianrized data as inputs to these approaches is appropriate for making an "apples to apples" comparison.

3_ Given that the proposed method is a DL-based approach, I believe it is important to provide some background on deep learning methods for single-cell RNA sequencing or spatial transcriptomics, or refer the reader to review papers on those topics.

Minor Comments:

1) The paragraph starting on line 39 provides valuable background on FISH and its limitations. I believe this section can be strengthened by adding a few references.

2) (Personal opinion) The word "tiny" (on page 2) seems a bit informal. I'd suggest replacing this word by "small" or "minute."

Point-by-point response

Dear Dr. Eldridge,

Thank you for considering our manuscript, “Predictive and robust gene selection for spatial transcriptomics” for review by *Nature Communications*, and for allowing us to submit a point-by-point response. We have incorporated all the feedback we received into a revised version of the manuscript, which we believe has addressed all the reviewers’ main concerns. We hope that you will find the revised paper significantly improved.

Decision on 07/19/2022

Thank you again for submitting your manuscript "Predictive and robust gene selection for spatial transcriptomics" to Nature Communications. We have now received reports from 3 reviewers and, after careful consideration, we have decided to invite a major revision of the manuscript.

As you will see from the reports copied below, the reviewers raise important concerns. We find that these concerns limit the strength of the study, and therefore we ask you to address them with additional work. Without substantial revisions, we will be unlikely to send the paper back to review.

If you feel that you are able to comprehensively address all of the reviewers’ concerns, please provide a point-by-point response to these comments along with your revision. Please show all changes in the manuscript text file with track changes or colour highlighting. If you are unable to address specific reviewer requests or find any points invalid, please explain why in the point-by-point response.

We would also like to thank the reviewers for their careful consideration of our manuscript and for their many suggestions for improvement. In response to the reviewers’ comments, we have made important changes that we feel address all of the reviewers’ concerns. Overall, the manuscript has greatly benefited from their feedback.

Here is a summary of the changes:

Experiments

- New plots quantifying cell type classification accuracy on per-type level (confusion matrix, recall by cell type)
- New plots quantifying expressed gene prediction accuracy on per-gene level
- New results quantifying PERSIST’s computational cost (run-time, memory usage)
- New baseline method, SMaSH (Nelson et al., 2022)
- New results testing marker genes from Tasic et al. (2018) for SSv4 cell type classification
- Quantified variance explained by cell types to contextualize explained variance results

- Improved demo notebooks in our PERSIST repository: now shows data downloading and pre-processing using ScanPy, integrates with AnnData, automatically installs dependencies

Writing

- Clarified that PERSIST’s memory usage can be managed via the minibatch size, which enables scaling to very large datasets
- Clarified motivation for supporting supervised and unsupervised gene selection (reference clusterings do not always exist and are not always reliable)
- Clarified motivation for including Seurat and Cell Ranger (widely used by practitioners)
- Clarified design of MERFISH experiment (we select genes and train imputation models using scRNA-seq data only, and MERFISH data cannot be used for training in practice)
- Added citations and clarifications about spatial transcriptomics/FISH limitations, and which types of studies can benefit most from PERSIST
- Added remarks to Methods about mitigating overfitting and existing deep learning methods in single-cell genomics (which are typically used to learn general non-linear representations)
- Expanded discussion section

Formatting

- Corrected numbering and improved ordering for supplementary figures
- Reflected individual data points on Fig. 3 bar charts
- Integrated footnotes into the main text

Below is a point-by-point response to the referee comments. The reviewers’ comments are in blue and italicized, and our replies are in black.

Reviewer #1:

In “Predictive and robust gene selection for spatial transcriptomics”, Covert and colleagues present a framework for gene panel selection that leverages single-cell sequencing datasets. Their machine learning approach employs an autoencoder with a mask layer to find gene panels that best predict the genome-wide expression profiles of single cells. The authors test this approach on two neuron-focused single-cell datasets and use the panels to classify cell types and estimate electrophysiological data. While some of the methodological ideas in this paper are interesting, the validation or performance estimate are unusually weak.

Major issues:

1) The authors do not test the autoencoder in external datasets and cannot gauge whether it is overfitting or overtraining. Recently, Grabsky and Irazarry highlighted the pitfalls of within

dataset learning of cell type annotations by comparing performance on external and internal withheld test cells (<https://doi.org/10.1093/biostatistics/kxac021>). They examined the effects of noise (incorrect labels), downsampled counts, and cells. At a minimum, the authors should similarly investigate the effects of within dataset evaluations, undertake cross-dataset evaluations, and significantly revise the manuscript accordingly.

Thank you for pointing out this issue. Overfitting is indeed an important challenge when using machine learning, and here we address this issue comprehensively:

- It would not be accurate to say that the manuscript does not test our approach on external datasets. Our experiment with MERFISH data (Fig. 5) uses data that is not only external, but collected using a different technology (MERFISH vs. SSV4). These results involve training with scRNA-seq data and evaluating with MERFISH data. PERSIST still outperforms the baseline methods, and moreover, its performance improves steadily with larger gene panels, so we do not see any evidence that overfitting is a significant concern.
- Regarding the idea of downsampling cell counts, such an investigation is provided in part by our experiment with the Sst sub-population (Fig. 2E-F). This experiment compares PERSIST's results when run with all neuronal cells (n=22,160) and with the smaller set of Sst cells (n=2,701). Although it was not the main aim of our experiment, the results demonstrate that PERSIST works well even with the considerably smaller population (roughly 12% of the full population size).

When running PERSIST, and indeed throughout the paper, we use standard approaches to mitigate the overfitting issue, such as training models that aren't too large and performing early stopping. Finally, while reporting results on a withheld test set is the accepted convention in the field (and we have carefully reported such test results in the original manuscript), it is also worth noting that the method we propose can be combined with arbitrary neural network regularization techniques. For example, if overfitting is a significant concern due to low cell counts, one can use common techniques like dropout, batch norm, or L1/L2 regularization.

To address the reviewer's concern, we have now 1) updated our manuscript to reflect parts of the above discussion in the Methods section (please see "Models and hyperparameters"), 2) added a remark explaining that once the gene panel has been selected, one can switch to using a non-neural network model to perform cell type classification, such as the approach presented in Grabski & Irizarry (2022), 3) cited the suggested Grabski & Irizarry (2022) paper in connection with the above remark (see line 635).

2) Evaluation of the other methods using the loss function created for the PERSIST framework is somewhat circular. Unlike the PERSIST model, these approaches were not explicitly designed to identify panels that explain the variance of the top 10,000 most variable genes using binarized expression. In essence, there is a question about the generalizability of the performance statistic (particularly when one model is optimized with respect to it, even if that is changed to be evaluated across datasets).

Thank you for raising this point, it's an understandable question about our evaluation. The performance metrics we chose reflect a panel's ability to summarize the genome-wide expression profile, and they are in some cases already used by biologists to determine the value of a given gene set. For this reason, we believe that they are natural to incorporate into the selection procedure with PERSIST. It's worth emphasizing that designing a gene selection method based on these criteria (explained variance, expressed gene predictability, and cell type classification) is non-trivial, so enabling this capability is a key contribution of our work: PERSIST can do so due to its use of deep learning with a feature selection layer and a flexible choice of the prediction target, but most methods do not offer this capability.

As for the comparisons between PERSIST and other methods, we agree that PERSIST is designed to have an advantage over methods based on unrelated selection criteria. However, it is crucial to point out that these baseline methods are popularly used to select genes, so they are not used out of context. Our results serve to highlight how well each selection approach aligns with different notions of gene panel quality. For example, the MutInfo method shares a similar selection criterion with PERSIST-Classification and provides competitive cell type classification accuracy. In contrast, Seurat yields relatively weak results according to the explained variance metric, reflecting that its ranking criterion is a poor surrogate for this notion of panel quality.

To address the reviewer's concern, we have now added a sentence highlighting that PERSIST is explicitly designed to optimize for the quality measures used by practitioners (see line 140):

“These methods span a range of selection criteria, but PERSIST is the first method that can be adapted to multiple experimental objectives relevant to practitioners, and that was designed specifically for transferability to spatial transcriptomics studies.”

3) The strong separation in the results and narrative between supervised and unsupervised approaches seems unnecessary to me, given that the goal is to identify gene panels and not annotate cell types. It is an expected result that cell-type annotations will help, if available. If they are not available, it likely makes sense to focus on cell-type discovery first. Alternatively, it would make sense to develop a better ensemble of use cases that plausibly map to unsupervised panel selection. In many ways this manuscript felt like a white paper on panel selection for a particular high-throughput experiment. A broader zone of application would be helpful.

Thank you for raising this point. We would like to begin by noting that, while cell type discovery is a century-old endeavor in the brain, essentially dating back to Cajal, there is still no consensus on the number and identity of brain cell types beyond the subclasses (e.g., Sst, VIP). Moreover, the current understanding in the field is that this is likely an application-dependent issue (e.g., there seem to be fewer functionally-relevant clusters compared to transcriptomically-different clusters). Therefore, it is very likely that neuroscientists will continue to work with different definitions of cell types even within a given brain region of a given species. This view is reflected in recent work, including the recent review paper “What is a cell type and how to define it” published in *Cell* (Zeng, 2022).

We emphasize that the two main use cases highlighted by the reviewer, supervised and unsupervised, are not artificially invented by us. Instead, we have carefully observed the needs of the field while developing our method. There are many cases where a reference clustering does not even exist (e.g., non-model animals), and many other cases where, despite the presence of a reference clustering, practitioners are interested in researching a new clustering scheme or using a different clustering scheme (e.g., a less detailed clustering).

Overall, the reviewer is certainly correct in observing that these two approaches (supervised vs. unsupervised) are methodologically similar, because our deep learning framework allows users to simply change the prediction target. Nevertheless, they address different practical needs in the field. To address the reviewer's concern, we have added a sentence to our manuscript clarifying the motivation for supporting gene selection in the absence of cell types (see line 118):

“While spatial transcriptomics studies often have specific goals like classifying cell types [3, 5, 7, 9], enabling PERSIST to operate in an unsupervised manner is important because reference cell type clusterings are not always available, consensus definitions of cell types are still evolving [37], and focusing on gene expression enables unbiased characterization of complex tissues and specific brain regions.”

4) Although the authors examine different levels of granularity, the results do not provide information about performance across cell types with varying proportions. This could be achieved by providing confusion matrices, micro, and macro averaged metrics.

Thank you for pointing this out, we agree that it's worth investigating cell type classification accuracy on the level of individual types. We added new figures showing these results, please see Supp. Figs. 7-8 on pages 23-24. Briefly, these show 1) confusion matrices for the 113 cell types in the SSv4 dataset, and 2) the recall (true positive rate) for each individual type. These results take a large amount of space, so we only show results for 32-gene panels selected by PERSIST (Supp. Fig. 7) and PERSIST-Classification (Supp. Fig. 8). It is clear from these plots that PERSIST-Classification offers a definitive improvement for panels of this size, and that while most cell types have high recall, several remain low. This provides further motivation to either use larger panels or to classify cells into more coarsely defined types (as shown in Fig. 3).

5) The authors mention that PERSIST requires a significant amount of memory and that it is difficult to run on a GPU. Deep learning models are well suited for GPU's and PERSIST would not be considered 'deep'. It is also odd that the demo notebook is set to use only 128mb of GPU memory. Additionally, the binarized input expression data should be relatively small. Given that the number of genes that can be spatially assayed is expected to increase, details on these computational limitations should be provided.

I question the usability of PERSIST given the computational limitations and the lack of integration with single-cell software libraries. In addition, as Table 1 shows, a significant amount of tuning was undertaken to customize PERSIST to various tasks. I attempted to get the provided demo

notebook running on my macbook pro but gave up after failing to load the dependencies. The average lab seeking to identify a panel will not have the expertise to apply PERSIST in its current state. For example, the demo notebook is full of hardcoded variables that require machine learning expertise to understand ('hidden=[128, 128]' and 'max_epochs=500'). Creating a well-documented Google Colab notebook might help in this regard.

Thank you for raising these concerns, usability is crucial and we have taken several steps to improve this aspect of our method. Overall, we want to emphasize that PERSIST requires minimal tuning, that its memory usage can be easily managed via the minibatch size, and that it is straightforward to integrate with popular packages like ScanPy.

Hyperparameters. Thank you for your attention to detail in noticing our hyperparameter table. We want to clarify that there was in fact very little hyperparameter tuning involved in our experiments. PERSIST's performance is not sensitive to the number of hidden layers or the layer width, and we did not compare different activation functions. To emphasize this, we re-ran several experiments so that PERSIST uses identical architectures across all experiments (see our updated Supp. Table 1). With this, only two hyperparameters now differ across datasets: the number of epochs and the minibatch size. It would be ideal to set these parameters automatically, and we will be sure to explore ways of doing so as we improve the PERSIST package over time.

Profiling computational cost. Following your recommendation and a similar request from R2, we added new results quantifying PERSIST's run-time and GPU memory usage (please see the Methods subsection "Memory and running time"). Briefly, we showed how the run-time and memory usage change as we vary 1) the number of candidate genes and 2) the size of the selected panel. We find that the memory scales linearly in both parameters when other variables are held constant, and that the run-time scales roughly linearly as well. Our manuscript originally pointed out that GPU memory usage can become high when the number of candidate genes and the panel size are both large, but after correcting a minor bug that increased memory usage, such issues now do not arise even with 10,000 candidates and a panel of 256 genes. This is reflected in Supp. Fig. 14, where we see that no setting we tested exceeds the memory capacity of a NVIDIA 2080 Ti GPU (which is not especially large by today's standards). In cases with even larger datasets, two simple options to manage memory usage are 1) reducing the minibatch size and 2) performing a preliminary filtering step to reduce the number of candidates.

Improving notebooks and usability. Following your suggestion, we added new notebooks to our repository that integrate common programming tools for single-cell genomics. The first notebook shows how to fetch one of our datasets and perform the pre-processing using ScanPy. The next two notebooks load data saved in the first notebook via an AnnData object, and use it to run PERSIST and PERSIST-Classification. As for other required libraries, we have added a full list of packages to our setup.py file, which can be used to automatically install these in a Python environment. We will also be sure to add more demo notebooks as we improve the package over time.

6) Although not ground truth, the cell-type labels from the MERFISH dataset should be used for both training and evaluation.

We believe there is a misunderstanding regarding our MERFISH experiment, which may be related to the point above that this is an external evaluation. Our goal here is to test whether each gene panel can predict cell-level properties in the MERFISH data, and crucially, the models used here are trained using only scRNA-seq data. Using MERFISH data during training would not carry any practical relevance, because such training data, and particularly the labels, are not available in real studies. Regarding the use of cell type labels, these are not part of our evaluation because the cell type hierarchy reflected in the MERFISH dataset is not identical to the one in our SSv4 dataset. This is why our evaluation focuses instead on imputing unobserved genes.

We apologize for the confusion here, and we think the most valuable improvement we can make is to better clarify the goal of this experiment and its setup. In the Results section “Binarization enables gene expression prediction with MERFISH data,” the opening paragraph has been revised as follows (see line 275):

“PERSIST can identify informative marker genes for a variety of experimental objectives, but our previous evaluations used only scRNA-seq data due to the challenge of providing an unbiased comparison via FISH studies conducted with multiple panels. Nevertheless, such cross-modal experiments represent an essential use case, which is applying predictive models trained using scRNA-seq to data collected from spatial transcriptomics studies. Here, it is important to verify that binarizing expression levels enables such models to transfer successfully between technologies, which is difficult to ascertain because accompanying annotations are seldom available for FISH datasets (e.g., ground truth cell type labels, or expression levels of genes that are not part of the FISH panel). To investigate this question, we therefore devised a multi-step *in silico* experiment using the SSv4 scRNA-seq dataset in combination with data from a recent, large-scale MERFISH study [9].”

7) The Discussion section, which currently consists of two paragraphs, should be expanded to mention scalability, likely sources of error, and future directions. In addition, the generalizability of PERSIST beyond neurons should be discussed as the application to other datasets with cell types with less expressed genes may limit accuracy.

In general, the novel approach taken by the authors for gene set selection may be of interest from others searching for optimal gene panels. However, a better case needs to be made given the limited accessibility and marginal performance gains in supervised settings.

Thank you for the suggestions. We have followed this advice and expanded our discussion to touch on several additional topics, including computational scalability. For example, we reference our results profiling PERSIST’s computational cost, we describe the extent of hyperparameter tuning currently required, and we mention that a future improvement might include setting any remaining hyperparameters (such as the number of training epochs) automatically.

Reviewer #2

Covert et al describe in this manuscript a novel approach to feature selection from single-cell data, primarily single-cell RNA sequencing (scRNA-seq), which they term PERSIST. In brief, PERSIST is a machine learning approach that aims to identify a user-defined number of genes that best capture the diversity of expression observed in scRNA-seq data or, in alternative versions, aims to capture the features that themselves best capture companion labels or measurements, such as the labels applied after cell-type identification. The authors place this algorithm in the broader context of the growing single-cell field by highlighting the role it could play in guiding the design of targeted transcriptomics measurements such as the modestly or highly multiplexed FISH assays, e.g. MERFISH, seqFISH, osmFISH.

The authors well describe the development of their algorithm and then benchmark it against other approaches for identifying informative gene sets from scRNA-seq, namely the methods for selecting highly variable genes provided in the Seurat and Cell Ranger packages as well as the GeneBasis algorithm. In an interesting aside, they very nicely show that the PERSIST framework can be used to capture genes that are most associated with electrophysiological properties of single cells measured in Patch-seq. Finally, they explore the ability of PERSIST to identify genes that can be leveraged to train imputation models to predict the expression of MERFISH measurements in the mouse cortex based on companion scRNA-seq measurements in the same region.

Overall, given the now wide and standard use of scRNA-seq and the still unmet need in those measurements to identify optimal and most-informative gene subsets in combination with the growing excitement and use of targeted image-based approaches to single-cell transcriptomics, I believe that the PERSIST approach provided here will be of interest to a range of scientists, and I support the publication of this manuscript.

However, I have several comments and concerns that I believe that the authors will want to consider in a revision of their manuscript. I list them below organized into major and minor concerns.

Major concerns:

1) The authors clearly motivate the PERSIST algorithm for the purpose of leveraging scRNA-seq to select features for multiplexed FISH assays. I completely agree that this application is an important one. Yet, most of their manuscript does not deal with FISH data, and, in the context, of the FISH data they do explore, they leverage PERSIST to identify a minimal set of genes that can be used to effectively co-embed scRNA-seq with MERFISH data in order to predict gene expression. Unfortunately, in no context do the authors test the ability of their algorithm to guide the design of FISH gene panels in a direct fashion. I do not fault the authors for this point, as its likely not possible to use their method to design panels that are then experimentally tested as the FISH-based assays are not yet widely available. However, perhaps the manuscript would be

stronger and easier for the average reader to understand if the authors could place greater emphasis on the other applications in which they validate PERSIST?

Thank you for raising this concern, it's an important point that we're happy to discuss. To clarify, the goal of our MERFISH experiment (see "Binarization enables gene expression prediction with MERFISH data") is to demonstrate the specific use case we envision: using PERSIST to select a gene panel, conducting a spatial transcriptomics study using that panel, and applying a predictive model trained using scRNA-seq data to the newly collected FISH data (e.g., for cell type classification). Evaluating PERSIST for this use case is challenging due to the lack of ground truth labels accompanying the FISH data, which is why our experiment uses an existing MERFISH dataset consisting of ~250 genes, and the prediction goal is imputing unobserved genes (rather than classifying cell types).

Based on your comment, we can see that the above point was not sufficiently well explained in the text. The opening of this section now reads as follows (see line 275):

"PERSIST can identify informative marker genes for a variety of experimental objectives, but our previous evaluations used only scRNA-seq data due to the challenge of providing an unbiased comparison via FISH studies conducted with multiple panels. Nevertheless, such cross-modal experiments represent an essential use case, which is applying predictive models trained using scRNA-seq to data collected from spatial transcriptomics studies. Here, it is important to verify that binarizing expression levels enables such models to transfer successfully between technologies, which is difficult to ascertain because accompanying annotations are seldom available for FISH datasets (e.g., ground truth cell type labels, or expression levels of genes that are not part of the FISH panel). To investigate this question, we therefore devised a multi-step *in silico* experiment using the SSv4 scRNA-seq dataset in combination with data from a recent, large-scale MERFISH study [9]."

As for why we did not conduct new spatial transcriptomics studies using the PERSIST panels, this is because it would not have helped test whether PERSIST panels are more effective than those selected by other methods. For this, we perhaps could have conducted experiments with panels of different sizes generated by each method. However, as you wrote in your comment, there are challenges here: 1) the cost of such experiments would have been prohibitive, 2) we would require concurrent measurement of a different kind of information that can be predicted given a gene panel (e.g., electrophysiological signature, cell type labels), which is typically not available, and 3) evaluating methods across different tissues may carry limited comparative significance. Overall, we thought carefully about the best way to provide an unbiased verification of PERSIST's utility, and we concluded that our MERFISH study was the best option available.

Finally, it's worth pointing out that the remainder of our experiments provide more standard evaluations using scRNA-seq data, but with an extra step to make them resemble the spatial transcriptomics use case: we binarize the input data for each predictive task. In the studies involving explained variance, expressed gene prediction and cell type classification, binarizing the input data simulates what we would do after conducting a FISH study, similar to our MERFISH

experiment, only the labels used here are those provided by the scRNA-seq dataset. We have attempted to clarify this in the revised version of the manuscript. The section containing the cell type classification experiments now begins as follows (see line 190):

“As another evaluation metric, we tested how accurately the gene panels selected by each method can classify cell types, which is a common goal of spatial transcriptomics studies [3, 5, 7, 9]. We utilized transcriptomic cell types defined via the original SSv4 and 10X datasets for our evaluation, and the classification accuracy with binarized input data simulates the accuracy in a subsequent FISH experiment.”

2) The authors benchmark the feature selection process of PERSIST against other methods for selecting informative gene sets, and this benchmarking is quite thorough. However, I have some concerns with the way in which they discuss the comparison between their algorithm and the methods used for selecting genes with Seurat and CellRanger. Critically, the goal of the feature selection algorithms in Seurat and CellRanger seems to me to be very different than that of PERSIST. These approaches aim to reduce the number of genes in subsequent calculations in order to reduce noise and the amount of computation. As such, these approaches are not looking to remove redundancy in the information carried by the genes within their final ‘highly variable’ set and, in many ways, they do not aim to extensively minimize the number of genes. By contrast, I suspect that implicit to the PERSIST approach is the removal of genes from the final set that carry redundant information as this ability is what would allow small targeted panels to carry information on the expression of the entire transcriptome. Thus, I suspect that—almost by design—CellRanger and Seurat should perform very poorly for small numbers of genes and using them as a benchmark in this context might not be as appropriate. I think that the authors should provide a more nuanced interpretation of what this comparison between PERSIST and Seurat/CellRanger implies about the performance benefits of their method.

Thank you for raising this point. We agree that when one closely examines the Seurat and Cell Ranger selection procedures, it is not surprising that they offer weak performance with very small gene panels: they are not specifically designed to select such small numbers of genes, and their ranking criteria cannot account for redundancy between genes. However, we believe our experiments are important to include because practitioners may not have the computational background to examine each method and determine a priori which is best for their use case. Indeed, these are popular tools (Seurat and ScanPy each have >1000 GitHub stars) that are actively being used by practitioners to choose informative gene panels for FISH studies. Therefore, they provide key benchmarks, and quantifying their performance via our quality metrics is a worthwhile result to highlight.

That said, we want to be careful to discuss the comparison in a correct way. We have thus added the following remark when introducing Seurat and Cell Ranger as baselines (see line 134):

“These methods are designed primarily to reduce computation, but we include them as comparisons because they are widely used by practitioners.”

3) Along these lines, identifying the most variable genes has not been the way in which others have leveraged scRNA-seq to guide the selection of markers for image-based spatial transcriptomics panels. A more common approach is to first cluster the scRNA-seq data, then to derive marker genes for each cluster, i.e. genes that are differentially expressed within that cluster, relative to all cells not within that cluster. Of course, this approach discards genes that capture inter-cluster variability. Nonetheless, given the central importance the authors have placed on the possibility of using PERSIST to identify the most informative genes for spatial transcriptomics panels, I think it is important to have at least a minor exploration of the overlap between the genes produced by PERSIST and marker genes derived from the scRNA-seq data in this fashion.

Thank you for bringing this up, it's true that practitioners often define marker genes after clustering the cell population in transcriptomic space. When the clusters also define cell types, such an approach can be understood as a supervised gene selection method, similar to PERSIST-Classification but with some shortcomings: most importantly, making selections based on differential expression does not account for groups of complementary or redundant genes, which limits their predictive power. Nonetheless, we agree that such approaches are worth considering in our manuscript.

As a simple exploration, we considered the set of marker genes identified by Tasic et al. (2018) for our SSv4 dataset. Among the 59 markers mentioned in the paper that overlap with our 10,000 genes, we found that only 2 were selected by PERSIST within its 64-gene panel, reflecting minimal overlap. Similarly, PERSIST-Classification only shares 2 genes within its 64-gene panel. We also tested these marker genes for their cell type classification accuracy, and the results show that these markers are relatively ineffective at differentiating cell types (see Fig. 3B). They underperform comparably sized panels selected by other supervised methods, likely for the reason mentioned above.

4) Given the central role of scalability in the usefulness of many computational methods, it would be very helpful to the average reader to have some quantification of the computational time and memory usage for the different examples explored in this manuscript. It is also important to have some comment on how resource use will scale as the number of genes/cells increases.

Thank you for pointing this out, we agree that it is worth profiling PERSIST's computational cost. We added new results to the Methods section performing this analysis, please see "Memory and running time" (line 583). Briefly, we showed how the running time and GPU memory usage change as we vary 1) the number of candidate genes and 2) the size of the selected panel. We find that the memory scales linearly in both parameters when other variables are held constant, and that the run-time scales roughly linearly as well. In addition, PERSIST can scale to large numbers of cells due to its minibatched training: the GPU memory usage does not depend on the number of cells, and the run-time is controlled by how one sets the number of training epochs.

5) I had a difficult time understanding how to interpret the performance of PERSIST (and the other methods) in terms of the explained variance (Fig 2A,C,E), as the fraction of the explained variance is low in all cases. On one hand, this observation may not be surprising as I would anticipate that

a sizeable fraction of the observed variance is due to the stochasticity of gene expression and detection. Yet, given the stated goal of PERSIST, could the authors provide some comment on how much variance they anticipate is meaningful biological variance?

Thank you for raising this point, we agree that this question deserves more attention in the text. We share the instinct that a sizeable fraction of the variance is due to stochasticity of gene expression and detection, but this is important to verify. To get a rough idea of how much noise exists in the data, we conducted a simple experiment: we partitioned each dataset by cell type, we calculated the remaining variance within each type, and we used this to estimate the amount of variance explained by knowing the cell type. The results confirmed our intuition about high noise levels: we found that knowing the cell type explains only 19% of the variance for the SSv4 data, and just 11% of the variance for the 10X data. These numbers are similar to what we observe in our experiments, although interestingly, the PERSIST panels are able to explain more variance than this given a sufficient number of genes (≥ 64).

Based on these results, we have updated the manuscript to include the following text (see line 157):

“This is due not only to the many factors of variation in the full expression profiles, but to high noise levels in the data. To verify this, we calculated the amount of variance explained by cell types in each dataset. We found that cell type labels explained just 19% of the variance in the SSv4 data and 11% in the 10X data, suggesting high intra-type variability due to stochasticity in gene expression and detection. Perhaps surprisingly, the PERSIST panels can explain more variance than the cell type identity given enough genes.”

6) I also had a difficult time understanding the metric they provide for the accuracy of their predictions. My central concern here is two-fold. First, the definition of this calculation in the methods portion of the manuscript is concise and technical and would benefit substantially from more explanation. Second, it strikes me that a more straightforward calculation of accuracy is both possible and, perhaps, more meaningful. Specifically, for each gene, could the authors calculate what fraction of the cells expressed that gene when they were predicted to express it, what fraction express that gene when they were predicted not to, and what fraction of cells don't express that gene when they were predicted to? Do these values depend on properties of the gene, such as its average expression? Overall, given the low capture efficiency and digital nature of scRNA-seq data (i.e. a high drop out rate), I am surprised that accuracy values can be so high. To illustrate my point, one could easily imagine that over 70% of the cells within a given cluster in single-cell RNA sequencing might have no copies of an RNA even when that gene is expressed on average at meaningful levels within that cluster. Again, given the stochastic nature of expression, I would anticipate that it is hard to make predictions of the expression of this gene that are correct 70% of the time or more. Arguments like this one motivate my suggestion that they authors explore more intuitive measures of correspondence between the measured and predicted values could be useful.

Thank you for raising this point. We have clarified this subject in the text by revising and expanding our description of the expressed gene accuracy metric, please see “Evaluation metrics” in the Methods section (line 648). We actually did what you suggested, which is simply calculating

the portion of the time that our model's predictions agree with the actual measurements, i.e., whether a gene was detected or not.

To follow up on your other questions, we have added additional plots quantifying the prediction accuracy on a per-gene level. In Supp. Fig. 5, we show these gene-level results and plot them against two gene characteristics: 1) the portion of cells in which the gene is expressed, and 2) the gene's mean expression level. The results show that the hardest genes to predict are those with moderate mean expression, or those which are expressed roughly half the time – which is natural and intuitive. Overall, we agree that predicting whether each gene will be expressed is challenging due to the stochasticity of expression, and this is reflected in our results. But the fact that the accuracy improves with larger panels reflects that each gene's expression and tendency to drop out is dependent on the expression level of other genes, and that some genes are more predictive than others.

More minor concerns:

1) I take some issue with the description of the state-of-the-art for multiplexed FISH assays. In particular, the authors highlight that even in the largest surveys these methods target less than 5% of the transcriptome and in more typical studies less than 0.2%. Yet, both MERFISH and seqFISH+ have published measurements of ~10,000 transcripts, which is far higher than 5% of the transcriptome. Moreover, relatively standard measurements now target several hundred genes if not ~1000, which is more than 0.2% of the transcriptome.

We thank the reviewer for bringing this point to our attention, and we agree that specialized FISH techniques with highly multiplexed readout methods can indeed probe far more than 5% of the transcriptome. We have added a sentence to the main text clarifying that PERSIST is most useful for routine FISH experiments in which compact sets of genes are probed, which is necessary in many experiments (see line 40):

“In routine FISH experiments, only a small fraction of the transcriptome is targeted [3, 6, 9, 12]; this is in part because the complexity and duration of FISH experiments increases sharply with the number of target genes, and also because highly specialized methods capable of probing thousands of genes are applicable only to thin tissue sections and cultured cells [7, 17].”

2) The numbering and labeling of the supplemental figures was difficult to follow and should be fixed. For example, supplemental figures are not labeled as such in the captions but only in reference in the text, and these figures are called out of order.

Thank you for pointing out these issues with our supplementary figures, we have attempted to address them. The captions are now formatted as “Supp. Figure X: Caption,” which is consistent with our references in the text (e.g., “Supp. Fig. X”). We have also reset the figure counter so that supplementary figures have numbers beginning from 1. Finally, we have reordered parts of the Methods section so that most supplementary figures are referenced in numerical order.

3) On page 10, the authors state that “this represents the first evidence, to our knowledge, that a predictive model trained using only scRNA-seq data can be applied successfully in a spatial transcriptomics study”. I strongly suggest the authors consider softening this statement because there are many methods that have demonstrated co-embedding of scRNA-seq and spatial transcriptomics data and that such co-embedding can be used for information transfer between these different data sets. Thus, the notion of combining these two types of data modalities via a variety of computational methods is not new.

Thank you for pointing this out, we did not intend to overlook other work combining these two data modalities. We have increased the specificity of our claim to highlight what we believe is new and most promising about these results. The revised version of this sentence is the following (see line 314):

“This represents the first quantitative evidence, to our knowledge, that a predictive model trained exclusively with scRNA-seq data can be transferred successfully to a subsequent spatial transcriptomics study.”

This updated version is intended to highlight the differences with prior work exploring co-embedding of the two modalities, but please do let us know if you would recommend further revision.

4) The authors state on page 17 that “it can be useful to consider only genes with high expression levels, because these transcripts are easier to identify in a FISH experiment.” However, for the vast majority of FISH techniques, especially the highly multiplexed methods, the opposite is true. It is typically far harder to address very highly expressed genes than lowly expressed genes.

Thank you for raising this point. We agree with the reviewer that for highly multiplexed methods, concerns related to optimal crowding necessitate the selection of lowly expressed rather than highly expressed genes. However, PERSIST isn't geared towards this class of FISH methods because greater numbers of genes can be probed (up to thousands), thereby removing the need to judiciously pick a compact set of highly informative genes. We performed the experiment in question to address other issues prevalent in many experimental settings, namely issues related to tissue quality, RNase contamination, or low-resolution detection methods. We have clarified this in the text by modifying the highlighted sentence which now reads as follows (see line 608):

“For example, when working with specimens of lower RNA quality (e.g. post-mortem human samples), it can be useful to consider only highly expressed genes that can be readily detected.”

Reviewer #3

The authors introduce PERSIST (predictive and robust gene selection for spatial transcriptomics), a DL-based feature selection method aiming to inform targeted spatial transcriptomics (ST). This

approach has tremendous potential in facilitating targeted ST protocols, such as FISH, given that a set of genes must be identified a priori.

Strengths:

- I believe the idea of binarizing the scRNAseq to account for the binary nature of FISH data and using a Gumble-Softmax distribution (initially used for developing categorical/binary VAEs) to optimize the model given the binary inputs is novel.

- Most non-linear dimensionality reduction approaches for scRNAseq (e.g. DCA) suffer from unexplainable latent spaces, which could result in limited utility based on the use case. I believe having dimensionality reduction defined as the number of genes that explain the most variance is appropriate for this work and the desired application.

- The results on scRNAseq reconstruction and cell-type classification based on gene panels are very interesting and significant. However, I do believe more appropriate comparisons are missing (more on this in the Major Comments/Questions Section).

- I found gene expression imputation with MERFISH data fascinating. Though I believe the feature selection comparisons are incomplete (detailed in the Major Comments/Questions Section), this experiment shows the potential of transferring information from scRNAseq to other modalities, which can be tremendous.

Major Comments

1) My main concern is the lack of comparison with existing supervised/unsupervised gene selection methods for scRNAseq, which have shown to accurately narrow down the gene space based on a desired downstream task. For example, some popular single-cell methods such as scVI [Lopez et. al] can be modified to narrow down the gene space based on the importance of the genes to a downstream task (in the case of scVI, this can be done if one uses a linearly-decoded VAE (LDVAE)). Another recent DL-based approach, called N-ACT (Heydari et al.), uses neural attention to identify the most important genes in optimizing an objective. It seems that the same framework can be leveraged to optimize the same loss function used in this work to reconstruct the scRNAseq data, and after training, the n most important genes can be selected and used for targeted ST.

The authors do not compare their methods to the mentioned approaches, nor do they provide justification of why such comparison would not be appropriate. I believe adding such comparisons/justifications would distinguish this work from the existing approaches and further strengthen the manuscript. I would appreciate learning more about the authors' response to this comment before further evaluating the feature selection results.

Thanks for raising this point, we of course want our experiments to provide comprehensive comparisons with state-of-the-art methods. We looked into the two methods you suggested, but for separate reasons we do not think they can be added to our paper:

- For scVI, it's true that the package offers a "linear decoder" functionality. However, this approach does not identify important input features, it instead helps identify how each latent dimension affects each output; to identify important predictors, it would be more useful to have a linear version of the encoder module. We could not find any indication that the LDVAE is a valid feature selection approach, so it seems like an unfairly weak comparison to include in our experiments. One alternative would be to integrate PERSIST's binary mask layer into scVI, which is an interesting direction for future work, but developing this method is beyond the scope of our current manuscript.
- For N-ACT, this is certainly an interesting method for post-hoc identification of important gene predictors using attention values, and we have added a citation to our paper (Ref. 23 in the revised manuscript). However, the method is currently unpublished and we were unable to get the code working on our own. Therefore, beyond citing the paper, we believe it may not be the highest priority to include among a list of common baselines.

In searching for state-of-the-art methods, we came across a different method that was recently published and that we had not originally considered in our experiments: SMaSH (Nelson et al., 2022), which trains a supervised classification model and identifies marker genes using feature importance scores. This method is thus related to N-ACT, but it is published and has code available online with demonstrations. We thus added SMaSH to experiments throughout our paper, and it can be seen that PERSIST offers improved performance in most metrics.

2) Could the authors justify why (or if) it is appropriate to use binarized data as inputs to Seurat and Cell Ranger? I understand the authors' desire for removing data processing as a possible explanation for PERSIST's better performance, but I am not convinced that using binarized data as inputs to these approaches is appropriate for making an "apples to apples" comparison.

Thanks for raising this point. Binarization is a key part of our proposal because it enables models to transfer to spatial transcriptomics data, as demonstrated in the section "Binarization enables gene expression prediction with MERFISH data." With this step in mind, PERSIST benefits from accounting for binarization while selecting genes. Several other methods were designed to operate on raw or normalized scRNA-seq data, which creates a degree of inconsistency with PERSIST, so our motivation in exploring the benefits of binarization was actually to provide a more fair (or apples-to-apples) comparison. Indeed, our manuscript shows results both with and without binarization for those methods (Fig. 2 and Supp. Fig. 4). Seurat and Cell Ranger are somewhat unique in that they become less effective with binarized data, so one option would have been to remove these results, but we thought it was best to include them for the sake of completeness.

To clarify that our aim was to provide a more fair comparison, we have added the following text to the manuscript (see line 163):

"Importantly, PERSIST binarizes gene expression levels during training whereas Seurat, Cell Ranger and GeneBasis all use either raw or logarithmized expression counts. This creates a degree

of inconsistency among methods, so we asked whether this pre-processing step could account for differences in performance.”

3) Given that the proposed method is a DL-based approach, I believe it is important to provide some background on deep learning methods for single-cell RNA sequencing or spatial transcriptomics, or refer the reader to review papers on those topics.

Thank you for the suggestion. We have added a sentence in the Methods section to highlight several popular deep learning methods in single-cell genomics. There are many possible papers to choose from, but we selected representation learning methods that we view as related to PERSIST (see line 402):

“Deep learning-based reconstruction models have become popular for extracting low-dimensional embeddings in single-cell genomics [28, 29, 53], but PERSIST is designed to select a precise number of genes rather than fitting a general non-linear embedding.”

The tools we cited here are scVI [28], DCA [29] and SAUCIE [53].

Minor Comments:

1) The paragraph starting on line 39 provides valuable background on FISH and its limitations. I believe this section can be strengthened by adding a few references.

As suggested, we have added citations to the statements made in this paragraph.

2) (Personal opinion) The word "tiny" (on page 2) seems a bit informal. I'd suggest replacing this word by "small" or "minute."

Thank you for pointing this out, we have changed this to “a small fraction of the transcriptome.”

Reviewers' Comments:

Reviewer #1:

Remarks to the Author:

I think the author have reasonably addressed my comments.

Reviewer #2:

Remarks to the Author:

In this manuscript, Covert and colleagues provide a revision to their original submission describing the PREDICT algorithm. The authors position the development of this deep-learning-based approach for feature selection and prediction from single-cell RNA sequencing data as solving an urgent demand for guidance on designing targeted panels for methods such as MERFISH, seqFISH, or other FISH-based spatial transcriptomics methods.

As I described in my original review, the topic is an important one and while I have some concerns with the method as described, I continue to support the eventual publication of the manuscript. Overall, the authors have provided a careful consideration of the points that both I and the other reviewers raised. Unfortunately, there is one major point that I feel has not been adequately addressed.

In my original review, I raised the concern that the CellRanger and Seurat 'highly-variable' gene selection approaches are not fair benchmarks. Simply put, CellRanger and Seurat select genes with no goal of capturing the maximal information in a minimal gene set. Rather these methods seek only to reduce the computational complexity of subsequent computational step with no inherent constraint on the number of genes that they define as 'highly variable.' Thus, forcing the selected genes produced by these methods to small sets of genes is inconsistent with the goals of those methods, and it should come as no surprise that these methods perform poorly when benchmarked for a task for which they were not designed.

I certainly appreciate that the authors have added a sentence aimed at addressing my concern--namely: "These methods are designed primarily to reduce computation, but we include them as comparisons because they are widely used by practitioners." Nonetheless, the heart of my concern is that since these methods were not designed for the task at hand, their poor performance is unsurprising. The authors are presenting benchmarks for which the expected performance is poor, and I think that this point needs to be made clearly—something this sentence does not do.

Moreover, I also have a second concern with this sentence. In particular, the assertion that these methods are 'widely used by practitioners.' I agree in some aspects with this assertion, as well as the expanded point in the response letter that these packages have received numerous stars on github. Indeed, these packages are widely used, and they are used by multiple FISH groups as they analyze scRNA-seq data to aid in the design of panels. Nonetheless, the wide use of these packages does not imply that their algorithms for identifying 'highly variable' genes are widely used for the selection of FISH probe sets. In fact, I don't know of a single instance in which a group has simply taken the topmost variable genes from this 'highly variable' gene set to create a FISH panel. The far more common usage of these packages is to use these highly variable genes to identify clusters, to then derive markers for these clusters, and then to use these markers to create, in part, the FISH panel.

I could still see utility in presenting the benchmarking they have done with CellRanger and Seurat in their manuscript, but I feel that these benchmarks need to be presented with a clear acknowledgement that the algorithms used to generate those genes sets were not designed nor are they often used for this purpose and, for this reason, it is not expected that they will perform comparable to methods specifically designed for the task at hand.

Reviewer #3:

Remarks to the Author:

I appreciate the authors' point-by-point rebuttal. I commend the authors for the additional experiments performed, and I believe the additional results (both in the main and supplementary sections) have improved the quality of the initial submission.

Although the authors did not specifically add the methods I had in mind, their reasons for choosing another state-of-the-art method are reasonable. The additional clarifications made in the manuscript also address my concerns about using binarized counts for Seurat and CellRanger. Lastly, the authors have added great references on FISH data, as well as adding citations to established tools for scRNAseq analyses. Therefore, I believe the authors have adequately addressed most of my concerns from the first draft, and the current version, in my opinion, is ready for publication.

Point-by-point response

Below is a point-by-point response to the referee comments. The reviewers' comments are in blue and italicized, and our replies are in black.

Reviewer #1:

I think the authors have reasonably addressed my comments.

Thank you very much for your feedback, we are happy to hear that your earlier comments have been resolved.

Reviewer #2:

In this manuscript, Covert and colleagues provide a revision to their original submission describing the PREDICT algorithm. The authors position the development of this deep-learning-based approach for feature selection and prediction from single-cell RNA sequencing data as solving an urgent demand for guidance on designing targeted panels for methods such as MERFISH, seqFISH, or other FISH-based spatial transcriptomics methods.

As I described in my original review, the topic is an important one and while I have some concerns with the method as described, I continue to support the eventual publication of the manuscript. Overall, the authors have provided a careful consideration of the points that both I and the other reviewers raised. Unfortunately, there is one major point that I feel has not been adequately addressed.

In my original review, I raised the concern that the CellRanger and Seurat 'highly-variable' gene selection approaches are not fair benchmarks. Simply put, CellRanger and Seurat select genes with no goal of capturing the maximal information in a minimal gene set. Rather these methods seek only to reduce the computational complexity of subsequent computational step with no inherent constraint on the number of genes that they define as 'highly variable.' Thus, forcing the selected genes produced by these methods to small sets of genes is inconsistent with the goals of those methods, and it should come as no surprise that these methods perform poorly when benchmarked for a task for which they were not designed.

I certainly appreciate that the authors have added a sentence aimed at addressing my concern--namely: "These methods are designed primarily to reduce computation, but we include them as comparisons because they are widely used by practitioners." Nonetheless, the heart of my concern is that since these methods were not designed for the task at hand, their poor performance is unsurprising. The authors are

presenting benchmarks for which the expected performance is poor, and I think that this point needs to be made clearly—something this sentence does not do.

Moreover, I also have a second concern with this sentence. In particular, the assertion that these methods are ‘widely used by practitioners.’ I agree in some aspects with this assertion, as well as the expanded point in the response letter that these packages have received numerous stars on github. Indeed, these packages are widely used, and they are used by multiple FISH groups as they analyze scRNA-seq data to aid in the design of panels. Nonetheless, the wide use of these packages does not imply that their algorithms for identifying ‘highly variable’ genes are widely used for the selection of FISH probe sets. In fact, I don’t know of a single instance in which a group has simply taken the topmost variable genes from this ‘highly variable’ gene set to create a FISH panel. The far more common usage of these packages is to use these highly variable genes to identify clusters, to then derive markers for these clusters, and then to use these markers to create, in part, the FISH panel.

I could still see utility in presenting the benchmarking they have done with CellRanger and Seurat in their manuscript, but I feel that these benchmarks need to be presented with a clear acknowledgement that the algorithms used to generate those genes sets were not designed nor are they often used for this purpose and, for this reason, it is not expected that they will perform comparable to methods specifically designed for the task at hand.

We are glad that you find our revisions helpful and support our manuscript’s publication, and we thank you for the many helpful suggestions in your previous comments. As for the remaining concern described above, we thank you for your attention to detail on this point. Like you said in your comments, these two methods (Seurat and Cell Ranger) were not originally intended to select small gene panels. They are used primarily to reduce dimensionality and computation, including for clustering analyses that sometimes inform marker gene panels. Using them to select gene panels is therefore somewhat unusual, but not entirely unreasonable: our experiments show that they sometimes match or exceed other baselines. Anecdotally, we are also aware of groups using these methods (at least initially) to select gene panels, although this has not been described in any publications we can cite.

However, overall, we agree that it would be surprising to see them perform on par with state-of-the-art methods. It may be best to view them as simple baselines, or perhaps even informative lower bounds (although they do occasionally outperform other baselines). As such, in light of your suggestion, we have revised the relevant section of the paper to include the following sentence (see line 134):

“These methods are designed primarily to reduce computation and inform clustering studies that help determine marker genes, but they are not intended to directly select gene panels; therefore, their performance is not expected to be competitive with methods designed explicitly for selecting small gene sets.”

Reviewer #3:

I appreciate the authors' point-by-point rebuttal. I commend the authors for the additional experiments performed, and I believe the additional results (both in the main and supplementary sections) have improved the quality of the initial submission.

Although the authors did not specifically add the methods I had in mind, their reasons for choosing another state-of-the-art method are reasonable. The additional clarifications made in the manuscript also address my concerns about using binarized counts for Seurat and CellRanger. Lastly, the authors have added great references on FISH data, as well as adding citations to established tools for scRNAseq analyses. Therefore, I believe the authors have adequately addressed most of my concerns from the first draft, and the current version, in my opinion, is ready for publication.

Thank you very much for your feedback. We thank you for your suggestions, which motivated several improvements to our writing and experiments.

Reviewers' Comments:

Reviewer #2:

Remarks to the Author:

I thank the authors for adding this one additional clarification. I think that their additional sentence is very nicely phrased and will help all readers understand exactly how to interpret these benchmarks.

Point-by-point response

Below is a point-by-point response to the referee comments. The reviewers' comments are in blue and italicized, and our replies are in black.

Reviewer #2:

I thank the authors for adding this one additional clarification. I think that their additional sentence is very nicely phrased and will help all readers understand exactly how to interpret these benchmarks.

Thank you again for your feedback, and we are glad to see that our revision was able to address your concern.